# Liver-target nanotechnology facilitates berberine to ameliorate cardio-metabolic diseases

Hui-Hui Guo[1,2], Chen-Lin Feng[1,2], Wen-Xuan Zhang [1,2], Zhi-Gang Luo[1,2], Hong-Juan Zhang[1], Ting-Ting Zhang[1], Chen Ma[1], Yun Zhan[1], Rui Li[1], Song Wu[1], Zeper Abliz[1], Cong Li[1], Xiao-Lin Li[1], Xiao-Lei Ma[1], Lu-Lu Wang[1], Wen-Sheng Zheng[1], Yan-Xing Han[1] & Jian-Dong Jiang[1]

Cardiovascular and metabolic disease (CMD) remains a main cause of premature death worldwide. Berberine (BBR), a lipid-lowering botanic compound with diversified potency against metabolic disorders, is a promising candidate for ameliorating CMD. The liver is the target of BBR so that liver-site accumulation could be important for fulfilling its therapeutic effect. In this study a rational designed micelle (CTA-Mic) consisting of α-tocopheryl hydrophobic core and on-site detachable polyethylene glycol-thiol shell is developed for effective liver deposition of BBR. The bio-distribution analysis proves that the accumulation of BBR in liver is increased by 248.8% assisted by micelles. Up-regulation of a range of energy-related genes is detectable in the HepG2 cells and in vivo. In the high fat diet-fed mice, BBR-CTA-Mic intervention remarkably improves metabolic profiles and reduces the formation of aortic arch plaque. Our results provide proof-of-concept for a liver-targeting strategy to ameliorate CMD using natural medicines facilitated by Nano-technology.

[1] State Key Laboratory of Bioactive Substance and Function of Natural Medicines, Beijing City Key Laboratory of Drug Delivery Technology and Novel Formulations, Institute of Materia Medica, Chinese Academy of Medical Sciences and Peking Union Medical College, Beijing 100050, China. [2] These authors contributed equally: Hui-Hui Guo, Chen-Lin Feng, Wen-Xuan Zhang, Zhi-Gang Luo. Correspondence and requests for materials should be addressed to L.-L. W. (email: wanglulu@imm.ac.cn) or to W.-S.Z. (email: zhengwensheng@imm.ac.cn) or to Y.-X.H. (email: hanyanxing@imm.ac.cn) or to J.-D.J. (email: jiang.jdong@163.com)

Cardiovascular and metabolic disease (CMD) is among the leading causes of morbidity and mortality both in developed and developing countries[1]. Hyperlipidemia, which can be associated with type 2 diabetes, obesity, non-alcoholic fatty liver disease, and inflammation, is an independent risk factor for CMD[2]. Berberine (BBR) is an isoquinoline derivative alkaloid presented in the root, rhizome, stem, fruit, and bark of various species of plants such as Coptis, Hydrastis, and Berberis. It is an approved nutraceutical substance for the treatment of hyperlipidemia in many countries[3], and has been suggested in the European Society of Cardiology & European Atherosclerosis Society (ESC/EAS) guideline as an alternative therapy to conventional statins for patients with hepatic damage or intolerant to statins[4]. In 2004, our team first discovered BBR a new lipid-lowering drug with novel mechanism (with respect to that of statins) against hyperlipidemia[5]. Since then, accumulating researches have verified its efficacy on lowering plasma cholesterol, LDL-cholesterol (LDL-c), and triglyceride (TG) levels through multiple target mechanisms, involving the production, excretion, and metabolism of plasma lipids[6]. Numerous clinical trials have confirmed its anti-hyperlipidemia effect[7] and many epidemiological and clinical data now support the tolerability and safety of BBR, especially in patients intolerant to statins[8]. In the past decade, this natural compound has been drawing attention and intensively studied for its benefits against various metabolic diseases, including diabetes and insulin resistance[9], liver disease[10], inflammation[11], obesity[12], and all of which are closely associated with CMD[13]. Moreover, a number of positive pleiotropic effects of BBR on the various stages of cardiovascular damage, such as regulating blood pressure[14], promoting cardiac function, improving endothelial function and arterial stiffness[15], anti-platelet, anti-prothrombotic[16], and anti-arrhythmia effects[17], have been reported. The available evidences, therefore, suggest that BBR is a promising candidate for the treatment of CMD.

Due to its poor bioavailability (<1%), a high dose of BBR (100–250 mg kg$^{-1}$ day$^{-1}$ for animals, 900–1500 mg day$^{-1}$ for humans) is needed for achieving therapeutic efficacy. Heavy dose-induced adverse effects, such as anorexia, stomach upset, diarrhea or constipation, can hinder its clinical use[18]. Numerous efforts have been made to increase its gut absorption among which permeation enhancers[19], P-glycoprotein (P-gp) inhibitors[20], and nanoparticle delivery systems[21] are the three major frontiers[22]. However, an active-site accumulation is presumably important for drugs to execute their therapeutic efficacy. Liver, the main organ in the regulation of whole-body energy homeostasis, is proved to be the main target of BBR[23]. Thus, we propose that a delivery system that can mediate selective liver-drug deposition might be a new strategy to enhance BBR's efficacy. Previous studies have suggested that the D-α-tocopheryl polyethylene glycol succinate micelles (TPGS-Mics) could increase BBR's permeability and bioavailability[20]. However, instability in physiological environment suppresses their effectiveness, as the Nano-carriers can't take the drug for active-site gathering[24]. In this study, a rational modified carrier (cross-linked D-α-tocopheryl polyethylene glycol-thiol succinamide micelle; CTA-Mic) was developed in an attempt to facilitate the effective liver accumulation of BBR. This novel system consists of a D-α-tocopheryl hydrophobic core and an on-site detachable cross-linked polyethylene glycol-thiol shell. The sturdy but liver-sensitive feature of the novel vector might maintain the integrity of the architecture during navigation, followed by a burst release of the payload after reaching the liver site. The system might pass through the gut wall in its intact form with few disturbances on the intestinal ecosystem. The trans-epithelial efficacy of BBR could be improved while the potential drug–drug interactions diminished. In circulation, the negatively charged micelles might protect BBR (positive change) from binding with proteins, as plasma proteins often carry negative charge. Mediated through micelle, the BBR uptake by hepatocyte was increased while the clearance of BBR by reticular-endothelial system decreased. Then, the redox- and enzyme-sensitive characteristics of CTA-Mic subsequently resulted in a burst release of BBR in liver cells. The degradation product D-α-tocopheryl succinate might assist the liver retention of BBR via the inhibition of P-gp efflux and cytochrome P450 (CYP450) elimination[25]. Moreover, degradation product D-α-tocopheryl, an antioxidant, might possess a synergistic effect with BBR in treating CMD[26]. Enhanced anti-CMD effects were expected at low-dose BBR application. The purpose of this study was to design and investigate the efficacy of BBR-CTA-Mic (50 mg kg$^{-1}$ day$^{-1}$ of BBR) on the treatment of CMD in the high fat diet (HFD)-fed mice.

Our results suggest that the designed BBR-CTA-Mic can effectively facilitate liver deposition and hepatocyte uptake of BBR. The in vivo study shows that intervention with the newly designed formula BBR-CTA-Mic (50 mg kg$^{-1}$ day$^{-1}$ of BBR) for 8 weeks can significantly stimulate the expression of a range of energy-related genes in the liver, decrease lipids and pro-inflammation cytokines in the plasma, thus ameliorate metabolic disorders and atherosclerosis in the HFD-fed mice. Conclusively, BBR-CTA-Mic might be a promising drug system for treating CMD.

## Results

**Synthesis and characterization of TPGTSA**. The chemical structure of D-α-tocopheryl polyethylene glycol-thiol succinamide (TPGTSA) was characterized by [1]H-NMR and [13]C-NMR spectrometry (500 Hz or 400 Hz; Varian, USA), Fourier transform infrared spectroscopy (Nicolet5700, USA) and MALDI-TOF mass spectrometer (Bruker Daltonics, Germany). (Fig. 1a and Supplementary Fig. 1)

**Preparation and characterization of BBR-CTA-Mic**. As shown in Fig. 1b (up), the CTA-Mics have a generally spherical morphology with an average particle size (PS) of 51.72 ± 1.82 nm (mean ± standard error mean (SEM), n = 5), polymer dispersity index (PDI) of 0.188 ± 0.038 (n = 5) and Zeta-potential (ZP) of-(18.16 ± 1.00) mV (n = 5). BBR loading did not influence the uniform micelle size with a minimal impact on the PS, PDI and ZP status (Fig. 1b, down), suggesting that most of the drug molecules were located in the inner space of the micelles. The morphological feature of BBR-CTA-Mic confirmed that most of BBR was distributed within the hydrophobic core and along the surfactant molecules in certain intermediate positions within the micelle's corona (Supplementary Fig. 2A). Previous research has suggested that TPGS could enhance BBR loading via the increased core and corona volumes, as well as the hydrophobic interactions of BBR alkaloid molecules with the aromatic chromane structure of D-α-tocopherol[21]. In this study, BBR-CTA-Mic demonstrated smaller average diameters, similar ZP and higher drug loading, with respect to BBR-TPGS-Mic did. These phenomena might be explained by using the similar but more robust material TPGTSA in the novel vector. For achieving the expected benefits from the nano-particulate carriers, keeping the integrity of particles until their arrival at the target site is necessary[27]. Then, the stability of BBR-CTA-Mic was evaluated by recording the variation in PS and PDI over time in simulated biological fluids. As shown in Supplementary Fig. 2B, C, after incubation with simulated physical fluids, no significant alterations have been observed in the BBR-CTA-Mic group, as compared to BBR-TPGS-Mic group. This result demonstrated that the cross-linked outer shell in combination with sturdy succinamide in the inner

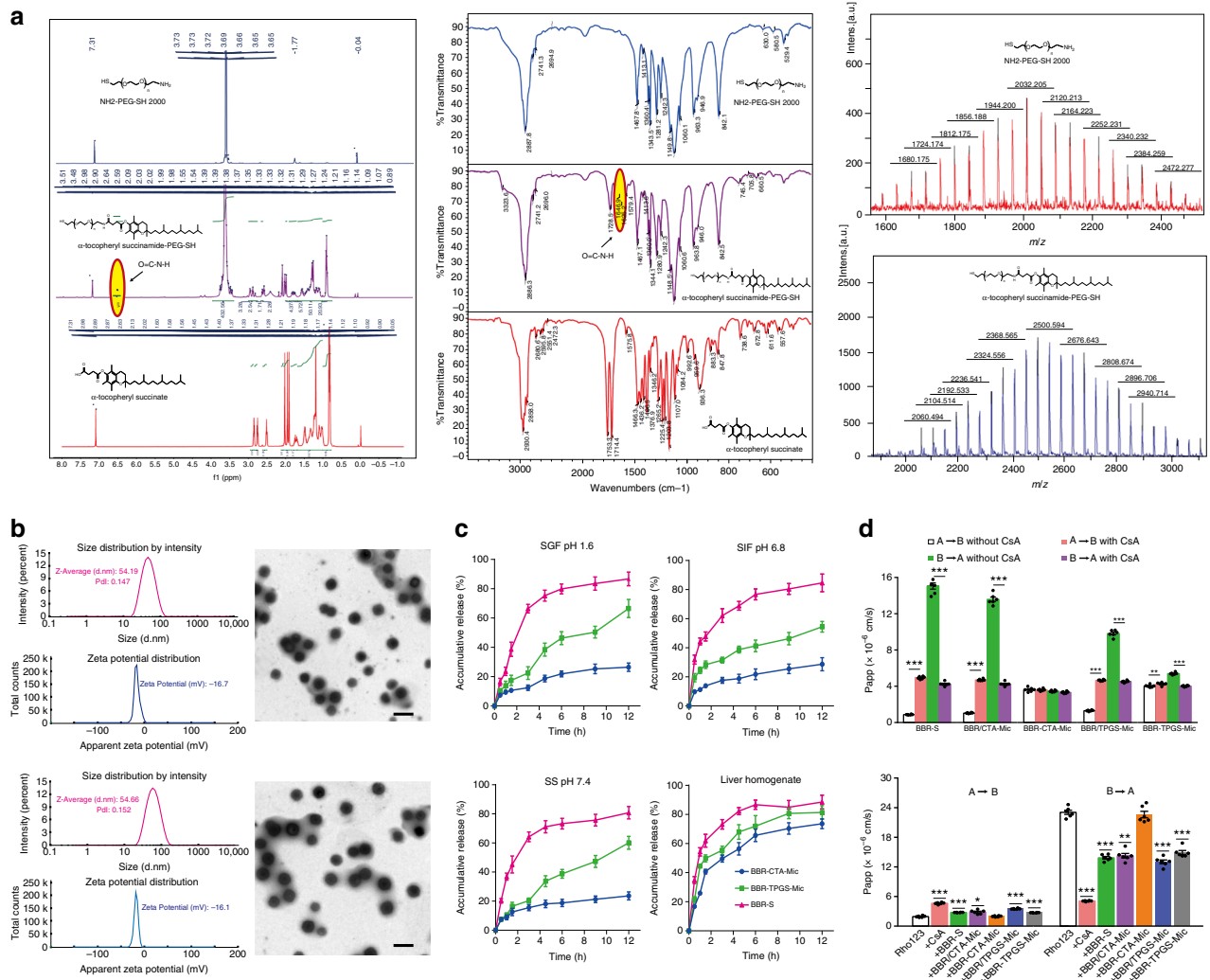

**Fig. 1** Characterization of TPGTSA and BBR-CTA-Mic. **a** The physical properties of TPGTSA. Left: Nuclear magnetic resonance hydrogen spectrum ($^1$H-NMR); middle: Flourier transformation infrared spectroscopy (FTIR); right: matrix-assisted laser desorption/ionization time of mass spectrometry (MALDI-TOF-MS). **b** Particle size, Zeta potential and morphology of empty CTA-Mics (up) and BBR-CTA-Mics (down). **c** Drug release study. The release of BBR from BBR-CTA-Mic and BBR-TPGS-Mic was carried out by dialysis method. BBR-S was used as control. 0.4 mL of BBR-S or BBR containing Mics was sealed in dialysis bags and submerged in 50 mL of fasting state simulated gastric fluid (SGF, pH 1.6), fasting state simulated intestinal fluid (SIF, 5 mM of bile salts, pH 6.8), simulated serum (SS, 20% FBS albumin, pH 7.4) and simulated liver environment (20% mice liver homogenate), respectively. Data are presented as mean ± SEM ($n = 5$). **d** Trans-epithelial transport study of BBR formulations in Caco-2 cell lines. Up: bidirectional apparent trans-epithelial permeability (Papp) of BBR contained formulations and the effect of CsA on the transport of different samples; down: the effect of different BBR containing formulations on Rho123 transportation. Data are presented as mean ± SEM ($n = 6$), *$p < 0.05$, **$p < 0.01$, ***$p < 0.001$, vs untreated control, $p$-values were calculated by unpaired two-sided Student's $t$-test. Scale bars, 100 nm (**b**)

core secured the stability of the BBR-CTA-Mics in physiological conditions. Consistent with the results from the stability analysis, more BBR was released from the BBR-TPGS-Mics under the condition of SGF, SIF or SS than that from the BBR-CTA-Mics (Fig. 1c). The BBR-CTA-Mics showed a slow-release phase over periods, irrespective of the pH or the composition of the test release media, reflecting greater drug incorporation stability due to the improved strength of the carriers. Interestingly, the burst release of BBR from BBR-CTA-Mics was detected upon exposure to 20% hepatic homogenate, demonstrating the sensitivity of BBR-CTA-Mics to the hepatocyte intracellular environment. The response of the BBR-CTA-Mics to liver environment was also confirmed by the destruction of the delivery nanoparticle via TEM imaging (Supplementary Fig. 2D). Possibly, the GSH-mediated cleavage of the disulfide bond on outer shell, together

with the amide enzyme-induced collapse of succinamide in the inner core, acted as triggers of vector disassembly, resulting in burst drug release.

**BBR-CTA-Mics assist drug absorption with few gut agitations.** The bidirectional apparent trans-epithelial permeability (Papp) and the corresponding ERs across Caco-2 cell monolayers are shown in Fig. 1d, Supplementary Table 1 and Supplementary Fig. 3. The results implied that efflux transporters were involved in BBR trans-epithelial transport, which was consistent with that reported in the previous research[28]. The results of the BBR plus CTA-Mic mixture (BBR/CTA-Mics) group were similar to that of the BBR-S group, implying that the addition of CTA-Mic did not influence the intestinal transportation of BBR. However, the Papp (A-B) of BBR was $3.63 \pm 0.23 \times 10^{-6}$ cm s$^{-1}$ ($n = 6$) and the Papp

(B-A) was $3.47 \pm 0.10 \times 10^{-6}$ cm s$^{-1}$ ($n = 6$) with an ER of 0.93 in BBR-CTA-Mics group, indicating that micelle encapsulation might increase the trans-epithelial effect of BBR by diminishing P-gp efflux. For the BBR plus TPGS-Mic mixture (BBR/TPGS-Mic) group and BBR-TPGS-Mic group, the Papp (A-B) value increased to $1.31 \pm 0.06$ and $4.02 \pm 0.15 \times 10^{-6}$ cm s$^{-1}$ while the Papp (B-A) decreased to $9.82 \pm 0.37$ and $5.40 \pm 0.13 \times 10^{-6}$ cm s$^{-1}$, respectively ($n = 6$). These changes might be explained by the instability of TPGS-Mic, since previous research suggested that TPGS could inhibit P-gp below its critical micelle concentration[29]. To further interpret the mechanism of trans-epithelial transport properties, the effect of P-gp inhibitor Cyclosporin A (CsA, 10 μM) on the transport of BBR containing samples were evaluated. After exposure to CsA, the Papp (A-B) of BBR-S increased while Papp (B-A) decreased, verified that P-gp involved in the efflux of BBR. A similar result was achieved for the BBR/CTA-Mic group. In contrast, exposure to the P-gp inhibitor did not influence the trans-epithelial transportation of BBR in the BBR-CTA-Mic group. Neither the Papp (A-B) nor Papp (B-A) was altered significantly. Notably, the Papp (A-B) was increased while the Papp (B-A) was decreased for the BBR/TPGS-Mic and BBR-TPGS-Mic groups, indicating the influence of P-gp efflux and further confirming the instability of BBR-TPGS-Mic. Next, the trans-epithelial transport of Rho123 (10 μM), a P-gp substrate reagent, was evaluated in different BBR formulations. As shown in Supplementary Fig. 3D and Fig. 1d (down), a reduction to 22.1% of B-A transportation of Rho123 was achieved by CsA addition, while no significant difference was seen after BBR-CTA-Mic addition. However, the addition of BBR/TPGS-Mic, BBR-S, BBR/CTA-Mic or BBR-TPGS-Mic was found to increase the absorption of Rho123 via P-gp inhibition, consistent with previous report[30]. Ultra-centrifugal-filter was used to test the integrity of micelle after transportation through the monolayer. As shown in the Supplementary Fig. 4, more than 50% of the BBR-CTA-Mics crossed the monolayer in their intact form, while almost all of the conventional BBR-TPGS-Mics collapsed during the transportation.

**BBR-CTA-Mics enhance drug accumulation in HepG2 cells.** After intestine absorption, liver is the first organ for BBR after a short travel from portal vein. Internalization of BBR-S, BBR/CTA-Mic, and BBR-CTA-Mic into hepatic cells was tested using confocal laser scanning microscopy (CLSM). As shown in Fig. 2a (left), after 3 h incubation, the cells treated with BBR-CTA-Mic displayed stronger green fluorescence of BBR as compared to the cells treated with BBR-S or BBR/CTA-Mic, indicating an increased cellular uptake of the BBR-CTA-Mic. The enhanced uptake of BBR-CTA-Mic by HepG2 cells was further demonstrated by flow cytometry (FCM) analysis (Fig. 2a, right). The mean fluorescence intensity (MFI) ratio of BBR-CTA-Mic (vs control) was 0.854- and 0.799-fold higher than that of BBR-S and BBR/CTA-Mic (Supplementary Fig. 5A). The accumulation of BBR in the HepG2 cells was further tested longitudinally. As shown in Fig. 2B and Supplementary Fig. 5B, C, after 1, 4, and 8 h incubation, the MFI ratios of BBR in cells treated with BBR-CTA-Mic were 0.534-, 0.674- and 1.214-fold higher than that in the BBR-S treated ones, demonstrating an improved drug penetration and accumulation of BBR in hepatocytes with encapsulation into the CTA-Mics. In addition, the cellular uptake of conventional BBR-TPGS-Mic was also tested in hepatocytes (Supplementary Fig. 5D) and the mechanism of endocytosis of BBR-CTA-Mic was investigated (Supplementary Fig. 6).

Another concern is the P-gp mediated efflux of BBR. Previous research indicated that P-gp efflux promoted the elimination of BBR in the liver, which counteracts their therapeutic effects.

Hence, we further examined BBR dispel by P-gp efflux pump using gene manipulating technology. The HepG2 cells were treated with P-gp siRNA to knock down the P-gp gene expression. As illustrated in Fig. 2C (left), in the BBR-S group, after 8 h dispel, the green fluorescence intensity of BBR in cells deprived of P-gp expression (treated with P-gp siRNA) was more obvious than that in cells with normal P-gp expression (treated with mismatch siRNA (mmRNA)), implying P-gp a main motive power for BBR efflux. Whereas, the green fluorescence intensity of BBR differed insignificantly in BBR-CTA-Mic treated cells, with or without P-gp expression, indicating BBR-CTA-Mic could reduce the P-gp-mediated BBR efflux. These results were confirmed by FCM analysis (Fig. 2C middle, Supplementary Fig. 7A) and agreed with the previous report that D-α-tocopheryl succinate could block P-gp-mediated drug efflux[21]. The increased intracellular uptake and decreased P-gp efflux of the encapsulated drug might contribute to the drug accumulation in hepatocyte cells. The efficiency of gene expression interference was verified by RT-PCR (Supplementary Fig. 7B), western blot (Supplementary Fig. 7C) and FCM analysis (Fig. 2C right, Supplementary Fig. 7D).

**BBR-CTA-Mics facilitated liver deposition of BBR.** A good retention inside the active-site is presumably important for medicines to execute their therapeutic efficacy[31]. Liver is the main target for BBR. In this study, the time-dependent bio-distribution of BBR in vivo was compared after the intragastric injection of BBR-S or BBR-CTA-Mic. The results tested by IVIS Imaging (Fig. 3A) and LC-MS/MS (Fig. 3b) indicated that BBR had a higher distribution in the liver, as compared to that in other organs. In the BBR-S group, BBR reached a maximum concentration ($C_{max}$) 2 h after treatment, followed by rapid decline; however, BBR-CTA-Mic not only increased the $C_{max}$ but also prolonged the retention time of BBR in the liver. BBR level in liver tissue was further examined. The results of fluorescent imaging (Fig. 3c) and air-flow assisted ionization mass spectrometry imaging (AFAI-MSI) (Fig. 3d) verified that BBR in the BBR-CTA-Mic group was maintained at a higher level and for a longer period of time in liver tissue, as compared with that in the BBR-S group. This elevated liver drug concentration in the BBR-CTA-Mic group could be due to the improved gut absorption and hepatocyte uptake assisted by CTA-Mic. It is worth noting that the elevation of BBR accumulation in the liver (+248.8%) was much higher than that in the plasma (+99.1%) and other main organs, clearly demonstrating that BBR-CTA-Mic could optimally assist BBR accumulation in the liver and thus improve its therapeutic effects. Liver cell distribution profile of BBR-CTA-Mic showed that the content of BBR in hepatocytes was improved by the CTA-Mic system, while the drug taken by liver non-parenchymal cells (such as kupffer cells, liver sinusoidal endothelial cells, monocytes, etc) was decreased or not significantly interfered by the delivery system. This effect might benefit the therapeutic effect of active BBR (Supplementary Fig. 8). In addition, the plasma BBR level in the mice treated with BBR-CTA-Mic was higher than that in the BBR-TPGS-Mic treated ones. (Supplementary Fig. 9)

**BBR-CTA-Mics activate metabolism-related genes in vitro.** The expression of a range of energy metabolism-related genes was analyzed in the HepG2 cells after treating with BBR containing formulations. As shown in Fig. 4a, simultaneous activation on InsR, p-AMPK and LDLR proteins was visualized following the treatment with BBR-CTA-Mic, indicating its gene regulatory effect. The results were verified by FCM analysis (Supplementary Fig. 10). Additionally, the mRNA level of *InsR*, *LDLR*, *AMPK*,

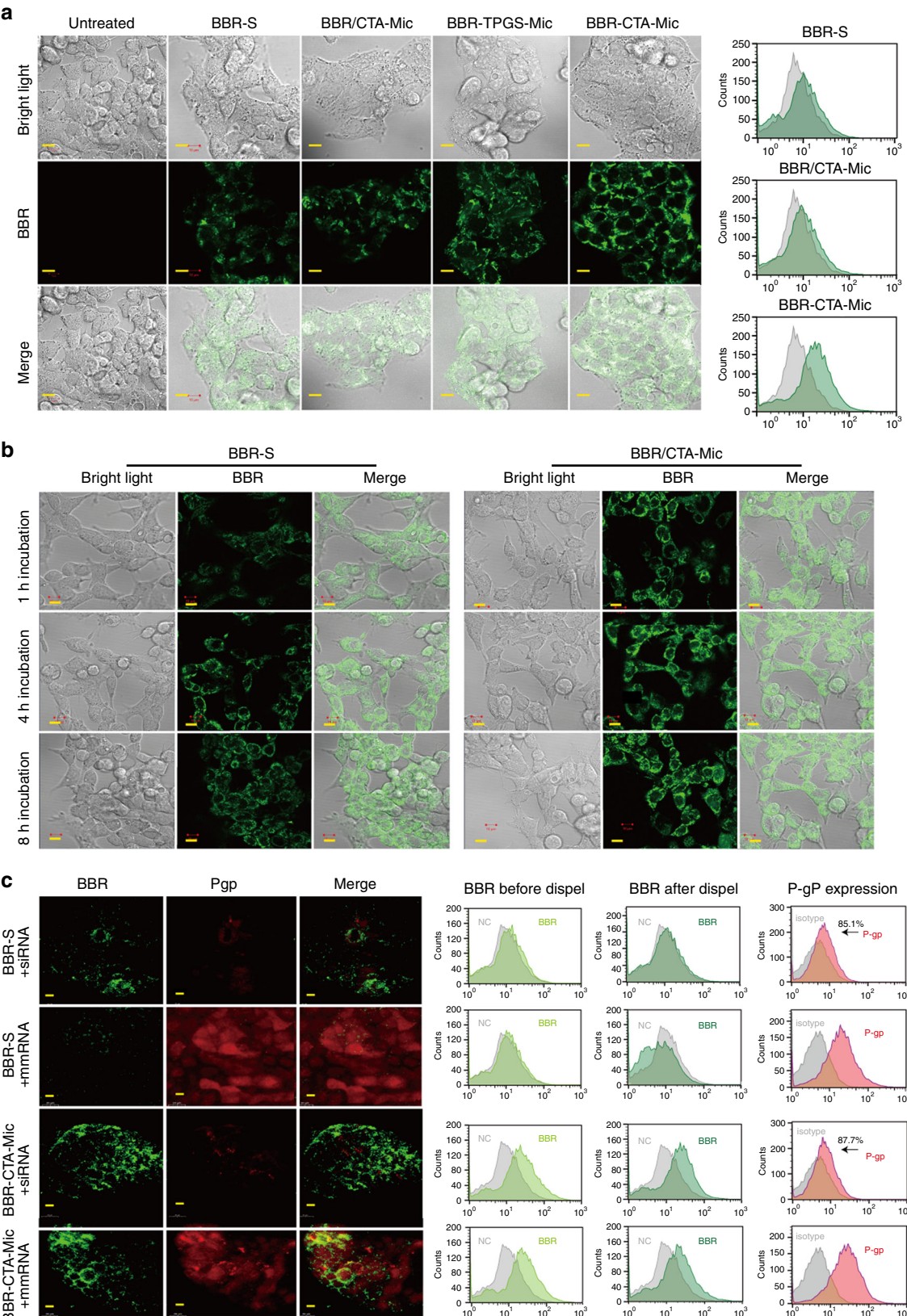

and *AKT* genes, as well as the protein level of InsR, *p*-InsR, LDLR, AMPK, *p*-AMPK, AKT, and *p*-AKT were tested for further interpretation of the gene modulating effect of BBR-CTA-Mic. As shown in Fig. 4b, the mRNA expression of *InsR*, *AMPK*, and

*LDLR* genes was elevated after adding BBR formulations. The protein expression of InsR, *p*-InsR, LDLR, AMPK, and *p*-AMPK was also increased after intervention with BBR formulations (Fig. 4c). The results are consistent with that in previous

**Fig. 2** Intracellular-uptake analysis. **a**. The cells were treated with various BBR formulations at an equivalent BBR concentration of 1 μg mL$^{-1}$ for 3 h, respectively, at 37 °C in 5% $CO_2$. Left: representative fluorescent images of BBR in HepG2 cells visualized using CLSM (Carl Zeiss, Germany); right: representative flow cytometry diagram of BBR in HepG2 cells. **b** The cells were treated with BBR-S or BBR-CTA-Mic for 1, 4, or 8 h, respectively. Representative fluorescent images of BBR in HepG2 cells visualized using CLSM. **c** HepG2 cells were pre-incubated with BBR-S or BBR-CTA-Mic accompanied with P-gp siRNA or mismatch siRNA (mm RNA) (50 nM) for 8 h. Cells were washed with PBS twice and incubated with fresh medium for another 4 h. Left: representative fluorescent images of BBR and P-gp in HepG2 cells visualized using C2t Nikon fluorescent microscope (Morrell, USA); middle: representative flow cytometry diagram of BBR deposition in HepG2 cells before and after dispel experiment. Right: representative flow cytometry diagram of P-gp expression in HepG2 cells before and after RNAi. The experiments were conducted in five times. Scale bars, 10 μm (**a–c**)

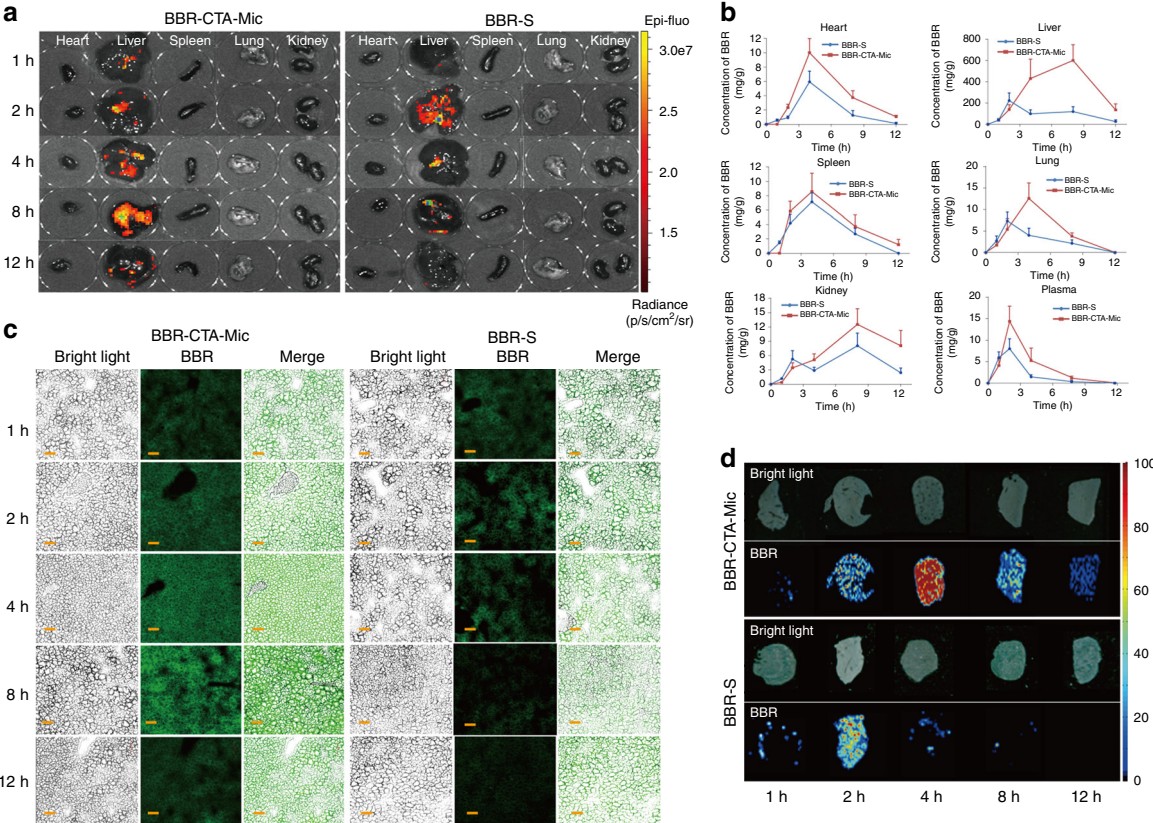

**Fig. 3** Bio-distribution Evaluation. C57BL/6J mice were administered with BBR-S or BBR-CTA-Mic (50 mg kg$^{-1}$ of BBR) via gavage injection. **a** At each predetermined time point, a group of five mice for each formulation were euthanized and blood (0.5 mL) were obtained from posterior orbital venous plexus to a heparinized tube and major organs (heart, liver, spleen, lung, and kidney) were harvested. Major organs were imaged using the IVIS imaging system at excitation/emissio n = 465/540 nm. (Heart, liver, spleen, lung, and kidney, from left to right). **b** Quantitative analysis of the distribution of BBR in C57BL/6J mice at different time points achieved via LC/MS/MS. (n = 5, mean ± SEM). Liver-focused evaluation was applied to further assay the hepatic cell accumulation of BBR in BBR-S or BBR-CTA-Mic group. The cryostat sections of liver sample were prepared by cutting the liver tissue into 4 μm. **c** Liver cell accumulation of BBR was visualized using CLSM LSM710 (Carl Zeiss, Germany) Ex = 365 nm; Em = 480 nm. **d** Liver accumulation of BBR was tested using ambient mass spectrometry imaging method. The experiments were performed on an air-flow-assisted desorption electrospray ionization (AFADESI)-MSI platform equipped with a Q-Orbitrap mass spectrometer. Scale bars, 50 μm (**c**)

reports[5, 6, 32]. The ratio of *p*-AMPK to AMPK and *p*-InsR to InsR was increased as well. BBR formulations did not increase the expression of AKT and *p*-AKT in the HepG2 cells, agreeable with previous findings that the AKT was activated by BBR only when insulin was present[33]. The BBR-CTA-Mic exhibited higher stimulating capacity on these genes, with respect to BBR-S and BBR/CTA-Mic when the BBR concentration was equal. The strengthened gene activating effect of BBR-CTA-Mic might be attributed to the improved drug liver deposition.

**BBR-CTA-Mics enhance the expression of energy-related genes in vivo**. Based on the substantial gene stimulating ability of BBR-CTA-Mic in vitro, its gene stimulating effects were further

evaluated in the liver tissues of the HFD-fed mice treated with the BBR formulations. As shown in Fig. 5a and Supplementary Fig. 11, the expression of *LDLR*, *p*-AMPK and *InsR* genes was significantly elevated in the liver of the BBR-CTA-Mic treated mice (BM), as compared to that in the untreated model controls (MC) or mice treated with empty micelle (EM). The increase of *InsR*, *AMPK*, and *AKT* genes was confirmed by RT-PCR (Fig. 5b); and the increased protein of LDLR, InsR, *p*-InsR, *p*-AMPK, and *p*-AKT was verified with western blot analysis (Fig. 5c). The ratio of *p*-AMPK to AMPK, *p*-InsR to InsR, and *p*-AKT to AKT was found significantly increased as well. The gene activating effect was also observed in the mice treated with BBR solution (BS), but the magnitude was lower than that in the BM-treated mice. No

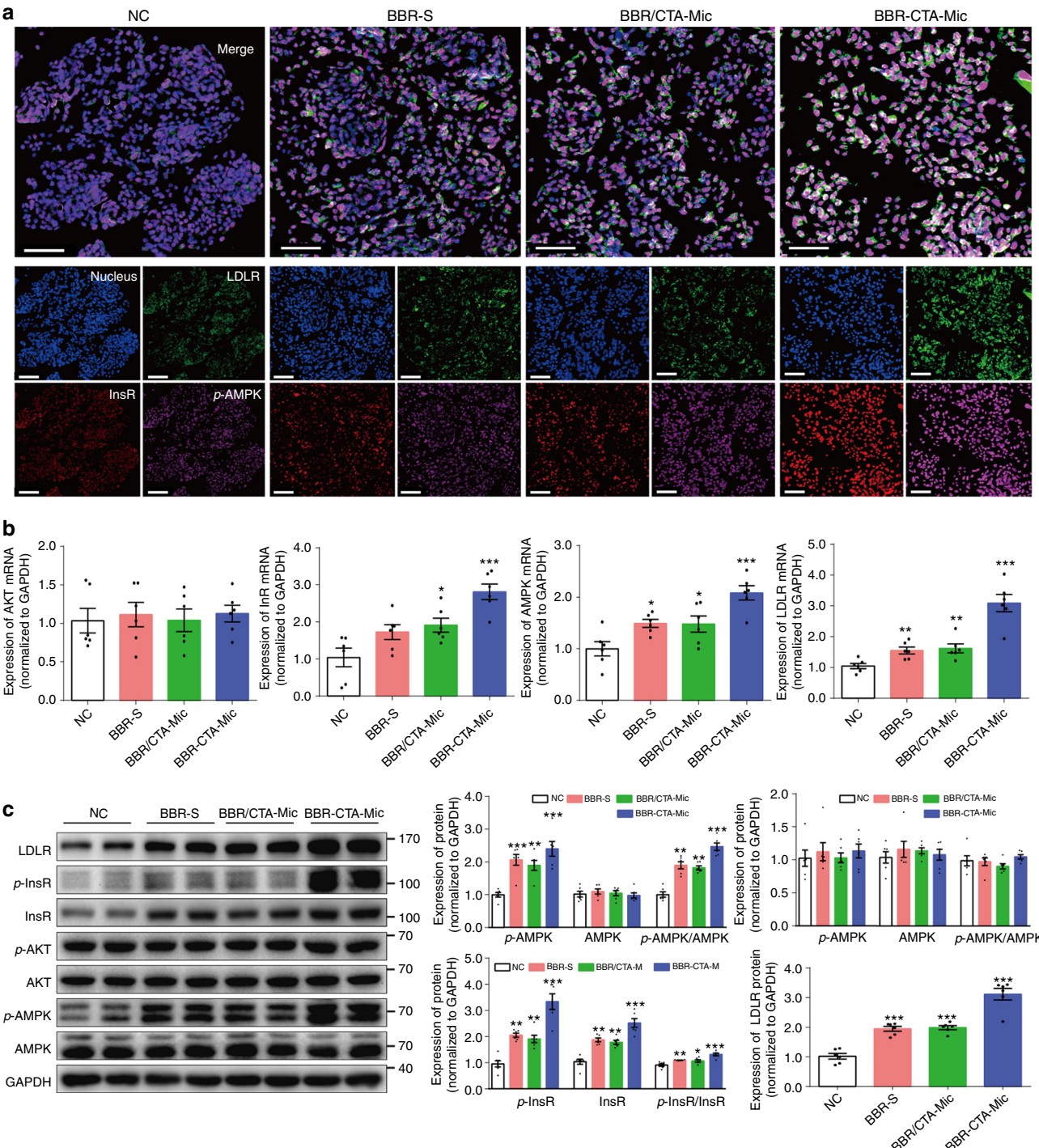

**Fig. 4** In vitro pharmacological effect. The HepG2 cells were treated with various BBR formulations at an equivalent BBR concentration of $1 \, \mu g \, mL^{-1}$ for 8 h at 37 °C in 5% $CO_2$. **a** The induction of $p$-AMPK (pink), InsR (red) and LDLR (green) protein expression by different BBR formulations was probed simultaneously with anti-InsR, LDLR and $p$-AMPK antibodies and visualized with C2t Nikon fluorescent microscope (Morrell, USA). **b** The mRNA expression of *AMPK*, *InsR*, *AKT*, and *LDLR* genes was evaluated by RT-PCR. The results were normalized to GAPDH. **c** The protein expression of AMPK, $p$-AMPK, InsR, $p$-InsR, AKT, $p$-AKT, and LDLR was tested using western blot analysis. The results were normalized to GAPDH as density ratio. Data are presented as mean ± SEM ($n = 6$), *$p < 0.05$, **$p < 0.01$, ***$p < 0.001$, vs NC group, $p$-values were calculated by unpaired two-sided Student's $t$-test. Scale bars, 100 μm (**a**)

stimulation effect was detected in the EM mice. These results demonstrated that BBR-CTA-Mic at low BBR dose (50 mg kg$^{-1}$ day$^{-1}$) could effectively induce expression of the genes central in energy modulation in vivo. In fact, the gene regulatory effect of BBR-CTA-Mic was in a dose-dependent fashion (Supplementary Fig. 12).

**BBR-CTA-Mics lower lipids and glucose level in HFD-fed mice.** As shown in Fig. 6a, the levels of plasma triglyceride (TG), cholesterol and LDL-c and hepatic TG in mice fed with normal chow diet group (NC group) were $0.54 \pm 0.11 \, mmol \, L^{-1}$, $3.43 \pm 0.28 \, mmol \, L^{-1}$, $0.40 \pm 0.10 \, mmol \, L^{-1}$ and $3.32 \pm 0.63 \, mg \, g^{-1}$, respectively ($n = 10$); and HFD feeding induced a significant

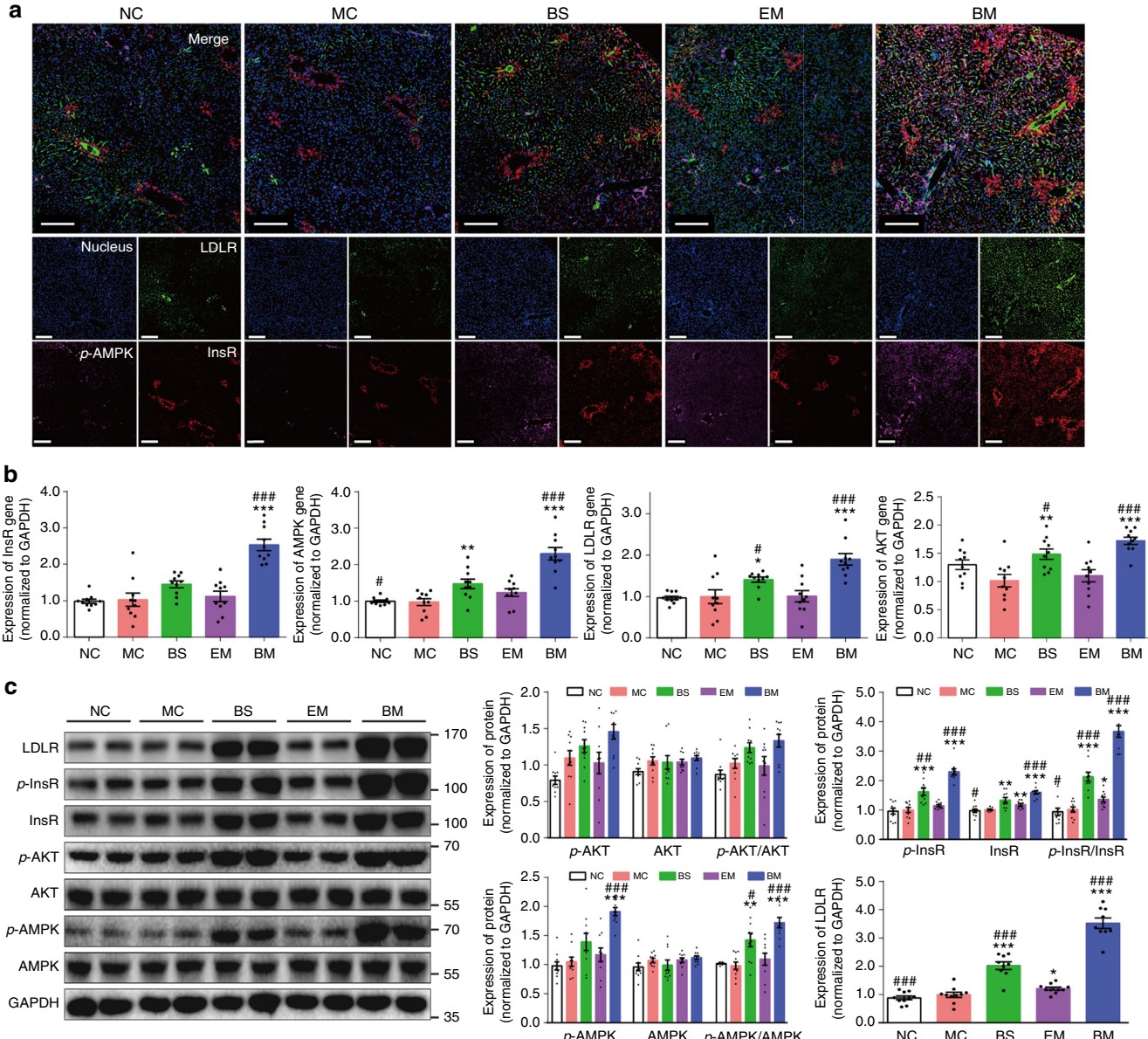

**Fig. 5** In vivo pharmacodynamics. C57BL/6J mice were feed with HFD for 8 weeks, followed with randomly allocated into four groups: model control group (MC), BBR-S group (BS, 50 mg kg$^{-1}$ of BBR), empty micelle group (EM, same as BBR micelles) or BBR-CTA-Mic group (BM, 50 mg kg$^{-1}$ of BBR), accompanied with HFD for another 8 weeks by gavage. Untreated mice fed with standard rodent diet (NC) were used as control. **a** Representative photographs of p-AMPK (pink), InsR (red) and LDLR (green) protein expression in liver tissue of different group mice were visualized using C2t Nikon fluorescent microscope (Morrell, USA) by probing with anti-InsR, LDLR and p-AMPK antibodies simultaneously. **b** The mRNA expression of *AMPK*, *InsR*, *AKT*, and *LDLR* genes was evaluated by RT-PCR. The results were normalized to GAPDH. **c** The protein expression of AMPK, p-AMPK, InsR, p-InsR, AKT, p-AKT, and LDLR was tested using western blot analysis. The results were normalized to GAPDH as density ratio. Data are presented as mean ± SEM ($n =$ 10), *$p < 0.05$, **$p < 0.01$, ***$p < 0.001$, vs mice in MC group; #$p < 0.05$, ##$p < 0.01$, ###$p < 0.001$, vs mice in EM group, p-values were calculated by unpaired two-sided Student's t-test. Scale bars, 200 μm (**a**)

elevation of these data (***$p < 0.001$, unpaired Student's t-test, two-sided), with respect to the NC group. BBR-CTA-Mic treatment significantly lowered these levels, compared with untreated MC group. The lipid-lowering effect was also observed in BS-treated mice, but the magnitude was lower than that in the BM-treated mice. For these lipid indications, no significant changes were found in mice treated with empty micelles ($p > 0.05$, EM vs MC). Moreover, similar modulation effect on blood glucose level was observed. The present study proved that significant lipid-lowering effect could be achieved with a low dose of BBR (50 mg

kg$^{-1}$ day$^{-1}$) through Nano-carrier facilitated liver-BBR accumulation.

**BBR-CTA-Mics lower weight-gain and adiposity in the HFD-fed mice.** Obesity is a high risk factor for CVD. Several studies attributed to BBR an anti-obesity effect as it could improve energy metabolism and inhibit adipogenesis[13, 16]. In this study, the body weight of HFD-fed mice, with or without drug treatment, was measured for 16 weeks. The liver, mesentery, and epididymal fat were harvested and weighed at the end of the experiment. As

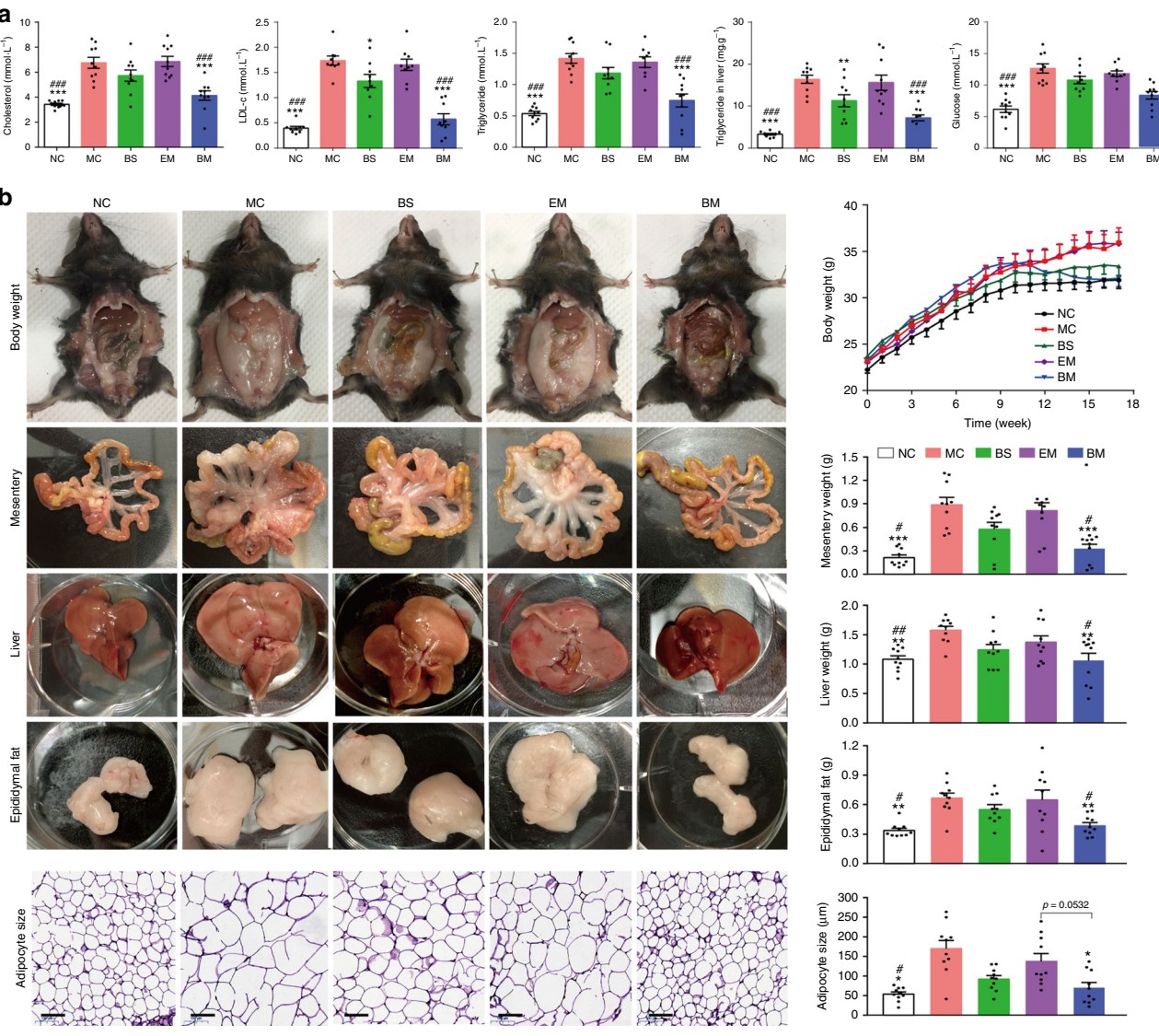

**Fig. 6** In vivo hyperlipidemia, body weight and adiposity analyses. HFD-fed C57BL/6J mice were treated with various BBR formulations by gavage. Untreated mice fed with HFD (MC group) and standard chow food (NC group) were used as control. **a** Biochemical analyses. Plasma TG, cholesterol, LDL-c, glucose, and hepatic TG were measured with enzymatic methods using an automatic biochemical analyzer. **b** Representative pictures and weight changes of whole body, mesentery and epididymal fat, liver tissue and size change of adipocyte. Data are presented as mean ± SEM ($n = 10$), *$p < 0.05$, **$p < 0.01$, ***$p < 0.001$, vs mice in MC group; #$p < 0.05$, ##$p < 0.01$, ###$p < 0.001$, vs mice in EM group, $p$-values were calculated by unpaired two-sided Student's $t$-test. Scale bars, 100 μm (**b**)

shown in Fig. 6b, mice in the MC group exhibited a remarkable weight-gain compared to those in NC group. Intervention with BBR-CTA-Mic for 8 weeks significantly reversed this increase of weight-gain in HFD-fed mice as compared to those untreated or treated with EM. As a result, mice in the BM group had a body weight similar to those in the NC group. Consistently, a significant elevation in the weight of liver, mesentery and epididymal fat was detected in the MC mice with respect to the NC mice. After 8 weeks medication, the BBR-CTA-Mic treatment substantially inhibited adiposity in the HFD-fed mice, while BS treatment resulted in a slight alleviation with no statistical significance. EM treatment expressed no effect on body weight or adiposity in the HFD-fed mice. Moreover, MC group showed a drastic increase in their adipocyte size relative to the NC mice. BBR-CTA-Mic medication restored the adipocyte size to that as the NC mice. The adipocyte size in the BS mice was smaller than that in the MC mice, but the difference was not significant ($p > 0.05$). In contrast, the tissue weights of the kidney, spleen, heart and

lung were not influenced by the HFD (Supplementary Fig. 13A). This study proved that BBR-CTA-Mic (50 mg kg$^{-1}$ day$^{-1}$ of BBR) intervention could successfully relieve HFD induced weight-gain and adiposity in mice.

**BBR-CTA-Mics alleviate fatty liver in the HFD-fed mice.** As seen in Fig. 7a, the photomicrograph of livers in the NC group indicated a normal hepatic structure with well-preserved cytoplasm and prominent nucleus. However, numerous spherical vacuoles of fat droplets were observed in the livers of MC mice, verified by Oil Red O staining (Fig. 7b). BBR-CTA-Mic intervention markedly decreased the hepatocyte vacuoles and lipid droplets (Supplementary Fig. 13B), reduced the plasma ALT levels in HFD-fed mice compared to absent of or EM treatment (Supplementary Fig. 13C, D). A mild decrease in the hepatocyte size, lipid droplet accumulation and ALT levels were detected in the BS mice but the magnitude was lower than that in the BM

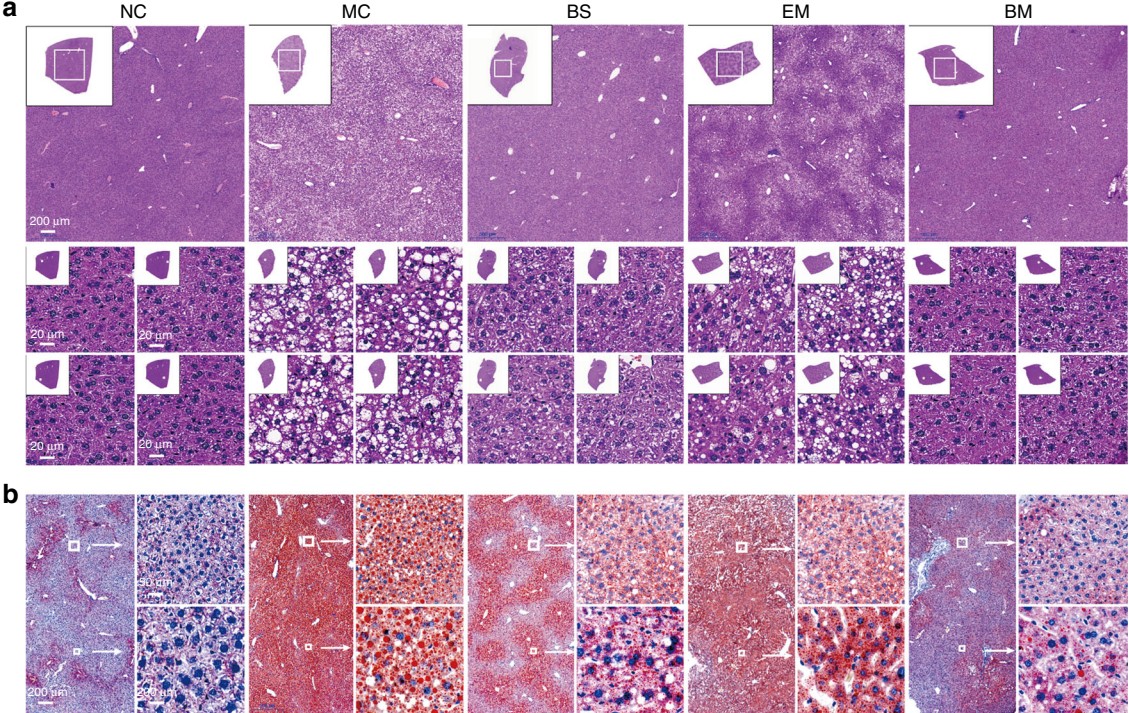

**Fig. 7** Fatty liver and hepatic injury evaluation. HFD-fed C57BL/6J mice were treated with various BBR formulations by gavage ($n = 10$ for each group). Untreated mice fed with HFD or standard chow food (NC group) were used as control. **a** Representative photographs of HE-stained liver sections. Insets contain images of whole liver tissue section. The regions of interest (ROI) are boxed in white, and their magnified images are shown at the right. **b** Representative photographs of oil Red O stained liver sections. The regions of interest (ROI) are boxed in white, and their magnified images are shown at the right

mice. No significant alteration was found in the EM group as compared to that in the MC group. In this study, we demonstrated that BBR-CTA-Mic ($50\,mg\,kg^{-1}\,day^{-1}$ of BBR) could reverse liver fat accumulation and liver injury induced by HFD.

**BBR-CTA-Mics ameliorate inflammation status in the HFD-fed mice.** Recent research has suggested that chronic inflammation is involved in the onset and development of CMD[34, 35]. Accordingly, pro-inflammatory cytokines have become the therapeutic target for cardiovascular diseases[36]. In the present study, the anti-inflammation effects of BBR formulations were examined. Ten pro-inflammatory factors: TNF-α, IL-1β, IL-6, IL-2, IL-10, IL-12, and IL-17, monocyte chemotactic protein 1 (MCP-1), MMP-9 and interferonγ (IFNγ) in plasma samples were tested. As shown in Fig. 8a, a substantial increase of TNF-α, IL-1β, IFNγ, and IL-6 production was detected in the MC mice, with respect to those in the NC mice. Treatment of the HFD-fed mice with BBR-CTA-Mic greatly inhibited these elevations. A mild decline of these cytokines was also observed in the BS mice but the magnitude was lower than that in the BM-treated mice. Accumulating evidence has suggested that the pro-inflammatory cytokines IL-6 and TNF-α are important in the response to atherosclerosis and valvar lesion[35] and are associated with endothelial dysfunction in patients with coronary artery disease or heart failure[37]. Then IL-6 and TNF-α expressed in liver and fat tissue were further evaluated. As shown in Fig. 8b, c, significant up-regulation of IL-6 and TNF-α level was visualized in liver and fat tissue of MC mice, and the elevated expression was down-regulated upon the treatment of BBR-CTA-Mic. It is noteworthy that the levels of IL-6 and TNF-α in BM-treated mice nearly returned to the levels comparable to that in NC mice. Although BS treatment tended to

reduce the hepatic and adipose IL-6 and TNF-α levels, the magnitude was lower than that of BM treatment. Interestingly, HFD-fed mice treated with EM showed decreased IL-6 and TNF-α in their fat and liver tissue, but not statistically significant. The mRNA and protein levels of IL-6 and TNF-α were quantitatively analyzed via RT-PCR and western blot (Fig. 8d, e). Consistent with the results of immunofluorescence, the levels of both cytokines in livers and fat tissues of the MC mice were significantly higher than those of NC mice. Upon treatment with BBR-CTA-Mic, the levels of IL-6 and TNF-α were significantly decreased, while the reduction by BBR-S or vector was at less significant level. This study showed that BBR-CTA-Mic ($50\,mg\,kg^{-1}\,day^{-1}$ of BBR) had an effective anti-inflammatory effect. Synergistic effect of vitamin E in the system might be attributable.

**BBR-CTA-Mic ameliorated atherosclerosis in the HFD mice.** Atherosclerosis formation within the entire aorta was detected. Representative Oil-red staining images of aortic arches and diverging blood vessels were recorded in all groups. As shown in Supplementary Fig. 14, the aortas of NC mice were almost devoid of fatty streaks while HFD feeding increased the lesion areas and induced increases in fatty streaks. BBR-CTA-Mic treatment lessened the development of these lesions. Our results showed that BBR-CTA-Mic intervention ($50\,mg\,kg^{-1}\,day^{-1}$ of BBR) could improve HFD induced metabolism disorders, lead to improvement of atherosclerosis.

**In vivo safety of BBR-CTA-Mic.** The long-term safety was investigated for 4 months to examine the biocompatibility in all groups. As shown in Supplementary Fig. 15, no obvious histological differences in major organs were found between the

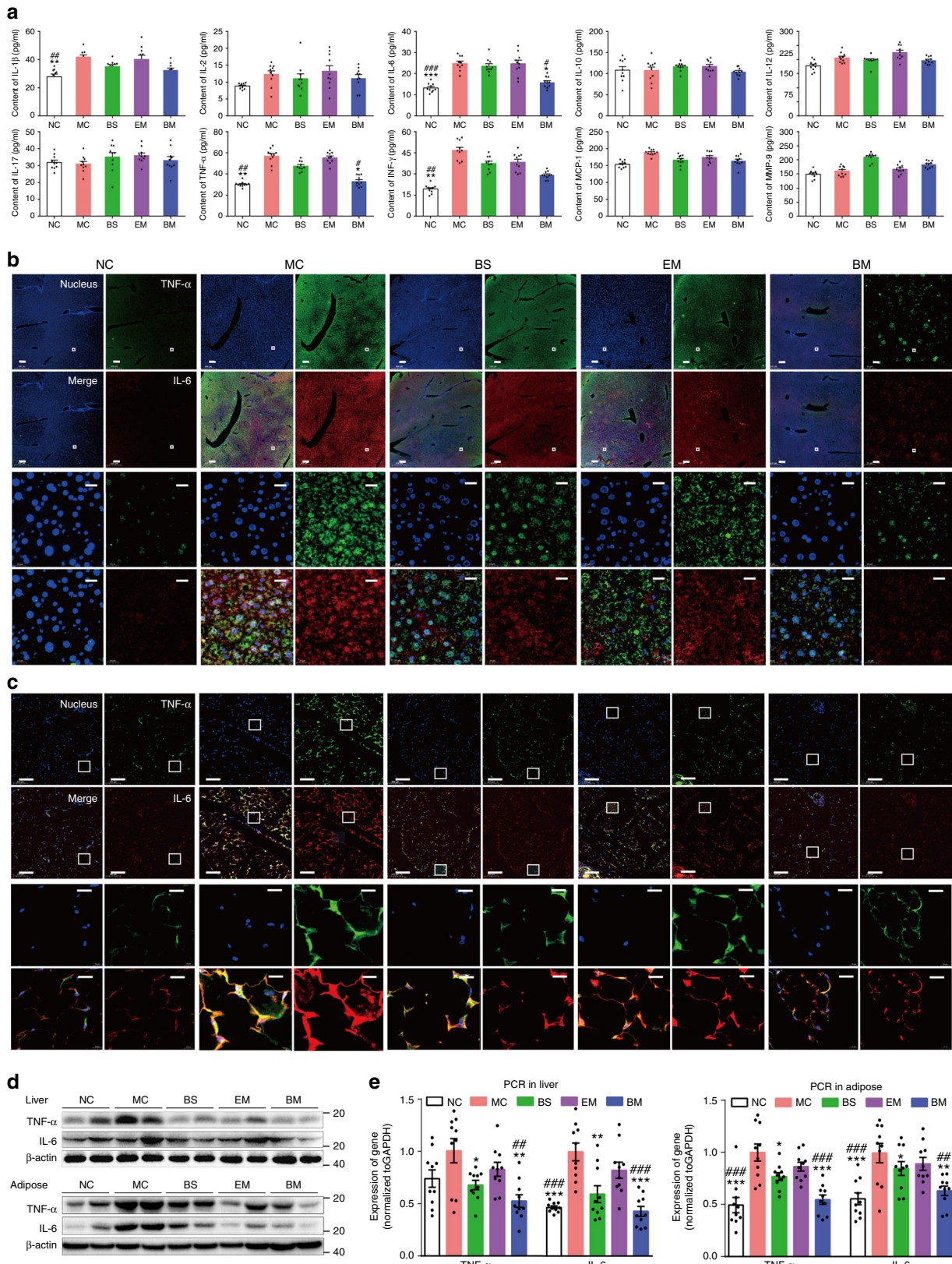

treated and untreated mice. No significant differences were detected between treatment group and the untreated control group in the plasma levels of ALT, AST, creatinine (CRE) and blood urea nitrogen (BUN), indicating a good tissue compatibility of the delivery system designed in this study.

## Discussion

This study developed a liver-targeted delivery system for BBR (BBR-CTA-Mic) to optimize its therapeutic efficacy on CMD.

Cardiovascular and metabolic disease (CMD) is among the leading causes of premature death worldwide. Hyperlipidemia,

**Fig. 8** Inflammation condition analysis. HFD-fed C57BL/6J mice treated with various BBR formulations by gavage. **a** Pro-inflammation cytokine levels in plasma. Following the termination of the experiment, blood samples were collected and used for the determination of plasma TNF-α, IL-1β, IL-2, IL-6, IL-10, IL-12, IL-17, MCP-1, MMP-9, and IFNγ levels by enzyme linked immunosorbent assay (ELISA) according to instruction of the manufacturer with a illuminometer at 490 nm. The tissue of epididymal fat and liver were harvested. **b** Representative photograph of immunofluorescent stained liver tissues for IL-6 (red) and TNF-α (green) visualized using C2t Nikon fluorescent microscope (Morrell, USA). The regions of interest (ROI) are boxed in white, and their magnified images are shown at the bottom. **c** Representative photographs of immunofluorescent stained adipose tissues for IL-6 (red) and TNF-a (green) visualized using C2t Nikon fluorescent microscope (Morrell, USA). The regions of interest (ROI) are boxed in white, and their magnified images are shown at the bottom. **d** The protein expression of IL-6 and TNF-α in liver tissue (up) and adipose (down) were evaluated by western blot. e The expression of IL-6 and TNF-α mRNA in liver tissue (left side) and adipose tissue (right side) were evaluated by RT-PCR. Data are presented as mean ± SEM ($n = 10$). *$p < 0.05$, **$p < 0.01$, ***$p < 0.001$, *vs* mice in MC group; #$p < 0.05$, ##$p < 0.01$, ###$p < 0.001$, vs mice in EM group, $p$ values were calculated by unpaired two-sided Student's $t$-test. Scale bars, 200 μm (**b**, **c** bottom left) and 20 μm (**b**, **c**, up right)

obesity, non-alcoholic fatty liver disease (NAFLD), and inflammation are independent contributors. BBR, a botanic compound, has been deemed as promising candidate for CMD treatment with its pleiotropic effect on these disorders. Atherosclerosis is the major cause of cardiovascular disease (CVD)[38]. Epidemiologic, clinical and experimental studies have shown energy metabolic disorders induce development of atherosclerosis[39, 40]. Numerous studies have suggested that BBR had beneficial effects in the prevention of arteriosclerosis[41, 42]. Chang et al. indicated that BBR could counteract HFD-provoked hyperlipidemia, thus ameliorating arteriosclerosis[43].

Due to the poor bioavailability of BBR, numerous efforts have been made to increase its gut absorption. As the next step, an active-site accumulation is presumably important for drugs to execute their therapeutic efficacy. Liver, the main organ in the regulation of whole-body energy homeostasis, is proved to be the main target of BBR[23]. Thus, we hypothesize that a delivery system that mediates selective liver-deposition would be good for BBR's action. In this study, a rational designed carrier (CTA-Mic) was developed aiming for a good BBR liver deposition.

It has been reported that the poor absorption of BBR might be attributable to its self-aggregation, poor permeability and P-gp-mediated efflux[44] in GI tract. Among these factors, P-gp mediated efflux is important. P-gp acts as a transport membrane pump and can effectively prevent exotoxin from entering circulation. BBR was reported to be a P-gp substrate, and P-gp efflux in the intestine has been suggested to play a role in its poor absorption[28]. On the other hand, BBR could alter the gastrointestinal absorption of other P-gp substrate drugs[30], causing a potential drug–drug interaction in poly-pharmacy. Vectors, that could improve permeability and reduce BBR efflux by P-gp due to the intact absorption of the wrapped drug, might provide a good solution to improve the oral bioavailability. Consistent with the results from the stability analysis, the novel BBR-CTA-Mics showed a slow-release phase over periods, irrespective of the pH or the composition of the test release media, reflecting good drug incorporation stability. The trans-epithelial transport study and micelles integrity test further proved that this delivery system could pass through the gut wall in its intact form. The permeability of BBR was improved while the potential drug–drug interactions diminished. Interestingly, the burst release of BBR from BBR-CTA-Mics was detected upon exposure to 20% hepatic homogenate, indicating a good sensitivity of BBR-CTA-Mics to the hepatocyte intracellular environment.

After intestine absorption, liver is the first organ for BBR after a short travel from portal vein. BBR in its original form could be taken and then removed by the liver reticular-endothelial system; and BBR in the hepatocytes could be metabolized by CYP450[45] or/and pumped out into blood circulation or bile by P-gp[44]. The cellular accumulation experiments showed an increased uptake and decreased elimination of BBR-CTA-Mic by hepatocytes. The

enhanced intracellular uptake of BBR-CTA-Mic may be ascribed to the distinct uptake mechanism of nanoparticles. Also, internalized micelle could enter cells carrying multiple drug molecules, generating an uptake process more efficient than the drug molecule diffusion[46]. The declined P-gp elimination agreed with the previous report that D-α-tocopheryl succinate, the degraded product of CTA-Mic in hepatocyte could block P-gp-mediated drug efflux. Furthermore, the in vivo distribution analysis verified that BBR in the BBR-CTA-Mic group was maintained at a higher level and for a longer period of time in liver tissue, as compared with that in the free BBR group. This elevated liver drug concentration in the BBR-CTA-Mic group could be due to the improved gut absorption and hepatocyte uptake assisted by CTA-Mic. It is worth noting that the elevation of BBR accumulation in the liver was much higher than that in the plasma and other main organs, clearly demonstrating that the BBR-CTA-Mic could optimally assist BBR accumulation in the liver and thus improve its therapeutic effects. The selective accumulation of BBR by CTA-Mic in the liver is probably because (1) the new formula increased the uptake of entrapped BBR by hepatocyte; (2) the catabolized products of the vector in hepatocytes could inhibit the activity of CYP450s and P-gp efflux[25, 47], thus slow-down the metabolism and excretion of BBR; and (3) it could evade the elimination of BBR by the liver reticular-endothelial system[48, 49]. Moreover, liver cell distribution profile of BBR-CTA-Mic showed that the content of BBR in hepatocytes was improved by the CTA-Mic system, while the drug taken by liver nonparenchymal cells (such as kupffer cells, liver sinusoidal endothelial cells, monocytes, etc) was decreased or not significantly interfered by the delivery system. This effect might benefit the therapeutic effect of active BBR.

It has been well established that the main mechanisms by which BBR exerts its regulatory effect on energy metabolism lay in its modulatory effect on LDLR, AMPK, InsR and AKT, among others. In 2004, Jiang's group revealed that BBR increased the expression of the *LDLR* gene at the post-transcriptional level via stabilizing *LDLR* mRNA, proposing one of the main mechanisms to explain the anti-hyperlipidemia effects of BBR[5]. InsR is an integral cell membrane glycoprotein and is important for the binding of insulin to target cells. The interaction between InsR and insulin causes a wide range of physiologic responses to maintain glucose homeostasis. Kong et al. proved in the HepG2 cells, as well as in the type 2 diabetes mellitus KK-Ay mice that BBR reduced insulin resistance through a protein kinase C (PKC)-dependent up-regulation of InsR expression[33]. As a major regulator of energy expenditure, AMPK has been shown to coordinate metabolic programs that increase energy expenditure and decrease energy storage[50]. Compiling results in vivo and ex vivo have shown that BBR could alleviate metabolic disorders, including obesity, insulin resistance, NAFLD and hyperlipidemia, by stimulating AMPK activity[51]. The present results

demonstrated that BBR-CTA-Mic at low BBR dose (50 mg kg$^{-1}$ day$^{-1}$) could effectively induce expression of the genes central in energy modulation in vivo and ameliorate CMD.

These data suggest that the designed BBR-CTA-Mic can effectively facilitate liver deposition and hepatocyte uptake of BBR. The in vivo study demonstrated that intervention with the newly designed formula BBR-CTA-Mic (50 mg kg$^{-1}$ day$^{-1}$ of BBR) for 8 weeks can significantly increase the expression of several energy-related genes in the liver, decrease lipids and pro-inflammation cytokines in the plasma, thus ameliorate metabolic disorders and atherosclerosis in HFD-fed mice. Conclusively, BBR-CTA-Mic might be a promising drug system for treating CMD.

## Methods

**Materials**. NH$_2$-(PEG)$_n$-SH (MW 2000 Da) was purchased from To Yong Bio (Shanghai, China). α-tocopherol and succinate were purchased from Sigma-Aldrich (St Louis, MO, USA). BBR was obtained from J&K Scientific Ltd (Beijing, China). Trypsin-EDTA (0.25%), cell culture media, penicillin/streptomycin, and FBS were obtained from Thermo Fisher Scientific (Waltham, USA). All other reagents were of analytical grade. All water used in the study was freshly double distilled.

The HepG2 and Caco-2 cell lines were obtained from the Cell Resource Center, Peking Union Medical College (which is the headquarter of National Infrastructure of Cell Line Resource, NSTI). The cell lines were checked free of mycoplasma contamination by PCR and culture. Their species origin was confirmed with PCR. The identity of the cell lines was authenticated with STR profiling (FBI, CODIS).

Male C57BL/6 J (6 weeks; 20–22 g) mice were purchased from Charles River (Beijing. China). The mice were housed in a temperature-controlled room 20 ± 1 °C on a 12 h light/dark cycle, with free access to water and food. Following an acclimation period of 1 week, the mice without lesion were included. The group size for each experiment is chosen considering the sufficiency to limit the variance within group to an acceptable scale and reflect the differences among groups. As for the randomization procedure, animals were firstly stratified by body weight and members in each layer were then allocated to different groups according to a random number table. All experimental procedures on animals were authorized by the ethics committee of the Institute of Materia Medica, Chinese Academy of Medical Sciences and Peking Union Medical College (Beijing, China). All diets of mice were purchased from Huafukang Bioscience Co. Inc (Beijing, China).

**Synthesis and characterization of TPGTSA**. TPGTSA was synthesized in two steps. The detailed synthetic methods are as follows. (1) α-Tocopherol (5 mmol) was used to react with succinate (5 mmol) in 20 mL anhydrous chloroform in the presence of DCC (10 mmol), and stirred constantly for 4 h at room temperature. The coarse product α-tocopheryl succinate (vitamin E succinate, VES) was obtained after the reaction solution was purified by silica column chromatography. (2) VES (100 mg, 0.179 mmol) and NH2-(PEG)n-SH (MW 2032, 377 mg, 0.180 mmol) were dissolved in chloroform, respectively. Then, the DCC (58.4 mg, 0.28 mmol) chloroform solution was added dropwise to the mixture of VES and NH$_2$-(PEG)$_n$-SH, and the solution was stirred at room temperature for 48 h. The solution was concentrated and purified by silica gel chromatography. The chemical structure of TPGTSA was characterized by $^1$H-NMR and $^{13}$C-NMR spectrometry (500 Hz; Varian, USA), Fourier transform infrared spectroscopy (Nicolet5700, USA) and MALDI-TOF mass spectrometer (Bruker Daltonics, Germany).

**Preparation and characterization of BBR-CTA-Mic**. 500 mg of TPGTSA copolymer was stirred with BBR previously dissolved in warmed ethanol at 20 mg mL$^{-1}$ concentration, followed by solvent evaporation, then, 4 mL of 4-(2-hydroxyethyl)−1-piperazine ethane sulfonic acid (HEPES) buffer (10 mM, pH 7.4) was added to the film, leading to formation of a clear micelle solution after 24 h of magnetic stirring. The final BBR-CTA-Mic was obtained after dialyzing against HEPES buffer (0.5% DMSO) and then washing with HEPES buffer using centrifugal filters. Empty micelle (EM) was prepared in the same way without BBR. BBR-TPGS-Mic was prepared as described before[20]. PS, PDI, and ZP of micelles were determined by dynamic light scattering using a Zetasizer (Nano ZS 90; Malvern Instruments, Malvern, UK). The morphology of micelles was evaluated by TEM (Hitachi H-7650, Hitachi, Tokyo, Japan) following negatively stained with 2% phosphotungstic acid. The encapsulation efficiency and drug loading content of BBR-CTA-Mic were determined using HPLC described before[20].

**In vitro BBR release and stability analysis of BBR-CTA-Mic**. The four organ mimic environments tested in this project were for stomach, intestine, blood, and liver, representing the physiological steps after BBR-CTA-Mic oral administration. The release test was designed to have BBR-CTA-Mic inside the bag and organ mimic environment outside the bag, in attempt to learn the release of BBR influenced by the organ environment. We anticipated that after exposure to the organ

mimic environment (outside bag), BBR-CTA-Mic interacts with the organ components that have proper size to penetrate into the bag, and might cause structure change of the micelle, resulting in BBR release to the outside bag environment. Briefly, 0.4 mL of BBR containing Mics (2 mg mL$^{-1}$ of BBR) was sealed in dialysis bags (Spectrum Laboratories Inc., molecular weight cut off: 14,000 Da, Los Angeles, USA). Then, the bags were submerged in 50 mL of different release media: fasting state simulated gastric fluid (SGF, pH 1.6), fasting state simulated intestinal fluid (SIF, pH 6.8 + 5 mM bile salts), simulated serum (pH 7.4 + 20% FBS) and 20% liver homogenate, respectively, with constant shaking at 200 rpm at 37 °C. At pre-determined time intervals, 0.5 mL of release mediums was sampled and replenished by the equal volume of fresh medium. The samples were filtered through 0.22 μm syringe membrane filter. The concentration of released BBR was analyzed by HPLC as described before[20]. Then, the stability of micelles was evaluated by incubating BBR formulations in these above simulated biological fluids for 2 h at 37 °C and the PS, PDI, and ZP were characterized before and after incubation. The physical stability of BBR-CTA-Mic was also evaluated to measure the changes in PS and ZP over five weeks of storage in refrigerated conditions (4 °C).

**Trans-epithelial transport study**. The Caco-2 cells were grown in minimum Eagle's medium (MEM) supplemented with 20% heat-inactivated FBS, 1% non-essential amino acids, 100 U mL$^{-1}$ penicillin and 100 U mL$^{-1}$ streptomycin at 37 °C in an atmosphere of 5% CO$_2$ and 90% relative humidity. Cell culture media were refreshed at 48 h intervals until confluence. Cells were seeded on the poly-carbonate filter inserts with a pore size of 0.4 μm and an area of 1.13 cm$^2$ (Corning Incorporated, NY, USA) at a density of 2 × 10$^5$ well$^{-1}$. After 21 days of culture, the tightness of cell monolayers was evaluated by testing trans-epithelial electrical resistance (TEER) with a Mill cell-ERS resistance instrument (Millipore, Billerica, MA, USA). Then, the inserts of TEER value higher than 600 Ω·cm$^2$ were washed twice and incubated with Hank's Balanced Salt Solution (HBSS) at 37 °C for 30 min. HBSS containing BBR-S, BBR/CTA-Mic or BBR-CTA-Mic were added either to the apical chamber (1.5 mL) or to the basolateral chamber (2.6 mL) to measure the polarized transport in absorptive and secretive directions. The corresponding opposite side was filled with HBSS as the receiving chamber at the same time. The HBSS added to basolateral side was adjusted to pH 7.4 and the apical side was adjusted to pH 6.8 for simulating the environment of circulation and intestinal mucosa. The experiments were performed at 37 °C in an incubator rotating at 50 rpm for 150 min. At different time points, Aliquots (100 μL) were taken from the receiver side and the same volume of fresh blank HBSS preheated at 37 °C was added immediately. The amounts of BBR in the media were assayed using a LC-MS/MS system described previously[20]. To test the effect of P-gp inhibitor Cyclosporin A (CsA), BBR containing formulations were added to the apical side or basolateral chamber in the presence of CsA (10 μM) in both chambers. Cells were incubated for 150 min and the concentrations of BBR in the receiving chamber were determined by LC-MS/MS analyses described previously[20]. To test the effect of different BBR formulations on P-gp substrate trans-epithelial trans-port, Rho123 (10 μM) was added to the apical or basolateral chamber, respectively, in the presence of BBR formulations at the BBR concentration of 20 μM. Cells were incubated for 150 min, and Rho123 concentrations in the receiving compartment were determined by an Envision Multilabel Reader 2104 (PerkinElmer, Waltham, USA). The TEER values were measured during the experiments to check the integrity of monolayers. Papp was calculated as described before[52].

At the end of the tans-epithelial transport experiment, 200 μL aliquots in the basolateral chamber were collected and transferred to Amicon® Ultra centrifugation filters (Merck Millipore, Billericia, MA, USA) with 3K Da cut-off, and centrifuged at 5000 × g for 15 min to detect the integrity of micelles[53]. The un-trapped BBR in the filtrate was evaluated by LC-MS/MS analysis described above. The integrity of the BBR containing system was evaluated by the formula: 1-(BBR in filtrate/whole transported BBR) %.

**In vivo bio-distribution of BBR-CTA-Mic**. BBR formulations were administered to C57BL/6J (6 weeks; 20–22 g) mice by gavage (50 mg kg$^{-1}$ of BBR). At each predetermined time point, a group of five mice for each formulation were euthanized and blood (0.5 mL) were obtained from posterior orbital venous plexus to a heparinized tube and major organs (heart, liver, spleen, lung and kidney) were harvested and imaged using an IVIS Spectrum Imaging System (PerkinElmer, Waltham, MA, USA) at ex/em = 465/540 nm. The distribution of BBR in various formulations was also analyzed by LC-MS/MS described previously[20]. Liver-focused evaluation was applied to further assay the hepatic cell accumulation of BBR. The cryostat section of liver sample was prepared by cutting the liver tissue into 4 μm. Hepatic cell accumulation of BBR was visualized using confocal laser scanning microscopy (CLSM) LSM710 (Zeiss, Oberkochen, Germany). Excitation was by Argon laser at 365 nm, and emission was captured through a 480 nm long pass filter, followed by ambient mass spectrometry imaging method. The experiments were performed and tested on an air-flow-assisted desorption electrospray ionization (AFADESI)-MSI platform equipped with a Q-Orbitrap mass spectrometer (Q Exactive, Thermo Scientific, Bremen, Germany)[54]. All the MSI data were acquired with full MS scan mode ranging from m/z 100 to 1000 in the positive mode, meanwhile the mass resolution was set at 70000.

**Cellular uptake and P-gp mediated efflux of BBR-CTA-Mic in HepG2 cells.** The cellular uptake of BBR of different formulations was examined using qualitative CLSM and quantitative flow cytometry analysis. HepG2 cells were seeded into 20 mm glass bottom cell culture dish (NEST Biotechnology, Wuxi, China) at $2 \times 10^5$ cells dish$^{-1}$ with MEM medium supplemented with 10% FBS for 24 h. Then, BBR containing formulations equivalent to 1 μg mL$^{-1}$ were added into the dishes and incubated for 3 h at 37 °C in 5% CO$_2$. After incubation, the cells were washed three times with PBS and immediately visualized using CLSM LSM710 (Carl Zeiss, Oberkochen, Germany). Excitation was set by Argon laser at 365 nm and emission was captured through a 480 nm long pass filter for BBR. Cells untreated were used as control. For flow cytometric experiment, HepG2 cells were grown in 6-well plates ($3 \times 10^5$ cells per well) and exposed to BBR containing formulations equivalent to 1 μg mL$^{-1}$ BBR. Subsequently, the cells were harvested, rinsed by PBS twice and centrifuged at $300 \times g$ for 5 min. Cells were suspended in PBS and immediately analyzed by BD FACS (BD Biosciences San Jose, CA, USA). Excitation was set 375 nm by a single 15 mW argon-ion laser beam and emission was collected through a 510 nm band pass filter.

To investigate the P-gp mediated efflux of BBR in BBR-S or BBR-CTA-Mic, HepG2 cells were exposed to 1 μg mL$^{-1}$ of BBR either encapsulated in micelles or freely dissolved in PBS accompanied with P-gp siRNA or mmRNA (50 nM, Santa Cruz, Dallas, TX USA) for 8 h to detect the BBR uptake using flow cytometry and CLSM. Also, the expression of P-gp was evaluated using western blot and RT-PCR.

**Mechanism of BBR-CTA-Mic endocytosis.** To investigated the endocytosis process of BBR-CAT-Mic, HepG2 cells were incubated with PBS (control), 10 μg mL$^{-1}$ chlorpromazine (a clathrin-mediated endocytosis inhibitor), or 10 μM nystatin (a caveolae-mediated endocytosis inhibitor) or 50 μM 5-(N-ethyl-N-iso-propyl) amiloride (a macropinocytosis inhibitor) respectively, for 30 min. Next, the cells were washed with PBS and further incubated with BBR-CAT-Mic at a concentration of 1 μg mL$^{-1}$ for 3 h. Then the BBR uptake was determined by flow cytometry.

**In vitro pharmacological effect.** To test the gene inspiring efficiency of the various BBR containing formulations, HepG2 cells were treated with different formulations (1 μg mL$^{-1}$ of BBR) for 8 h to evaluate the expression of energy-related genes using RT-PCR, western blot, flow cytometry, and triple-immunofluorescent staining analysis. Untreated cells were used as the control.

**In vivo pharmacological evaluation.** Male C57BL/6J (6 weeks; 20–22 g) mice were housed in a temperature -controlled room 20 ± 1 °C on a 12 h light/dark cycle, with free access to water and food. Following an acclimation period of 1 week, the mice without lesion were included and randomized into normal chow diet mice (NC group; $n = 10$) or a high fat diet mice (60% kcal fats, 20% kcal carbohydrates and 20% kcal proteins, HFD group; $n = 40$). Following 8 weeks, animals fed with HFD were randomly allocated into four groups (10 animals each group): model control group (MC group, untreated), BBR solution group (BS, 50 mg kg$^{-1}$ day$^{-1}$ of BBR), empty micelles group (EM, same as BBR micelles), and BBR-CTA-Mic group (BM, 50 mg kg$^{-1}$ day$^{-1}$ of BBR) for another 8 weeks by gavage. During the study, the animals were weighed weekly. At the end of the experiment, the mice were anesthetized after 12 h fasting period. Plasma was collected for biochemical and cytokine analyses. The tissues of mesentery, epididymis fat, liver, heart, spleen, lung, and kidney were harvested and weighed. All tissues were divided into two parts, one was immediately immersed in liquid nitrogen and stored at −80 °C for further analysis, and the remaining samples were fixed with 10% formalin for histological analysis.

**In vivo safety analysis of BBR-CTA-Mic.** The long-term safety was investigated for 4 months after drug administration to examine the biocompatibility of BBR formulations. C57BL/6J (6 weeks; 20–22 g) mice were received an intragastric dose of various BBR formulations (50 mg kg$^{-1}$ day$^{-1}$of BBR, $n = 5$) for 4 months. Untreated C57BL/6J mice ($n = 5$) were included as controls. The mice were sacrificed at termination. The blood was collected and analyzed for AST, ALT, CRE, and BUN to assess hepatic, heart and renal toxicity, respectively. The main organs (heart, liver, spleen, lung, and kidney) were processed for H.E. staining.

**Serum biochemical and cytokine analysis.** Plasma cholesterol, TG, LDL-c, glucose, ALT, AST were measured with commercially available kits (Biosino, China) using a TOSHIBA automatic biochemical analyzer (TOSHIBA Ltd., Tokyo, Japan) according to the manufacturer's protocol. The plasma levels of the cytokines TNF-α, IL-1β, IL-6, IL-2, IL-10, IL-12, IL-17, MCP-1, MMP-9, IFN-y were measured by enzyme linked immunosorbent assay (ELISA) according to instruction of the manufacturer (R&D Systems, MN, USA), using a illuminometer (Awareness Technology Inc., Palm City, FL, USA).

**Histological staining.** Tissue samples embedded in paraffin wax were sectioned into 4 μm thicknesses and stained with H.E. for photo-microscopic observation. Hepatic fat accumulation was evaluated by Oil Red O staining. The cryostat sections of liver were stained with 0.1% Oil Red to detect lipid droplets.

Photomicrographs were taken with a Nikon Eclipse E600 microscope (Nikon Corporation, Tokyo, Japan). For atherosclerotic lesion assessment, Vessels of aorta arch attached to thoracic aorta were isolated and fixed in 10% paraformaldehyde. Then, Oil Red staining of vessels was performed to analyze atherosclerotic lesions. The photographs were taken with Canon Digital Single Lens Reflex. Image-Pro-Plus 6.0 was used to qualify red O oil staining.

**Hepatic TG content determination.** 100 mg of liver tissue was extracted with a mixture of chloroform and methanol (2:1, v/v). The mixture was homogenized for 30 s and exposed to gentle agitation overnight at 4 °C to separate into 2 phases, and centrifuged at $655 \times g$ for 10 min at 4 °C. An aliquot of the organic phase was evaporated and re-suspended in ethanol. Total triglyceride concentrations of liver tissue were then measured by commercially TG kit (Biosino, China) according to manufacturer's instructions and TG content was expressed as Nano moles per gram wet liver weight.

**RT-PCR analysis.** Total RNA of hepG2 cells, liver and adipose tissue was extracted using TRIzol® Plus RNA Purification Kit (Invitrogen, CA, USA) and the RT-PCR experiment was conducted following the process described before[24]. The PCR primer sequences of related genes were in Supplementary Table 2.

**Western blot analysis.** Total protein from hepG2 cells, liver or adipose tissue were obtained to perform western blot and targeted proteins were probed with the indicated primary antibodies: LDLR (1:1000, 10785-1-AP), P-gp (1:1000, 22336-1-AP), IL-6 (1:1000, 21865-1-AP), TNF-α (1:1000, 60291-1-Ig), GAPDH (1:1000, 10494-1-AP) (Proteintech, USA); p-AMPK (1:1000, ab133448), AMPK (1:1000, ab80039), p-InsR (1:1000, ab60946) (Abcam, UK); p-AKT (1:2000, #4060), AKT (1:2000, #2920), β-actin (1:1000, #4970) (Cell Signaling, USA) or InsR (1:200, sc-57342, Santa Cruz, USA). The matching horseradish peroxidase (HRP) conjugated secondary antibodies (1:5000, #7076, #7074, Cell Signaling, USA) were used to evaluate the protein expression. The results were tested with ChemiDoc XRS electrophoretic imaging system (Bio-Rad Laboratories, Berkeley, USA) and normalized to GAPDH or β-actin.

**Flow cytometry analysis.** For P-gp mediated efflux of BBR-CTA-Mic, hepG2 cells fixed with cold ethanol were collected and stained with PE-conjugated mouse anti-human P-gp antibody (1:50, ab93590, Abcam, UK) or isotype control antibody (1:50, #61656, Cell Signaling, USA), and measured using BD FACS Calibur (BD Biosciences, CA, USA). Excitation was by a single 15 mW argon-ion laser beam (488 nm for P-gp or 375 nm for BBR, respectively). Emission was collected through 585 nm band pass filter for P-gp, or 510 nm for BBR, respectively.

The expressions of LDLR and p-AMPK proteins in hepG2 cells and hepatocyte from C57 mice were determined by staining with Alexa Fluor 488-conjugated rabbit LDLR monoclonal antibody (1:500, ab196377, Abcam, UK) or isotype control antibody (1:500, ab199091, Abcam, UK), or rabbit p-AMPK monoclonal antibody (1:50, ab23875, Abcam, UK) or isotype control (1:50, ab172730, Abcam, UK) combined with Alexa Fluor 647-conjugated anti-rabbit IgG (1:2000, ab150079, Abcam, UK). The InsR protein expression in hepG2 cells was determined by incubating with InsR monoclonal mouse antibody (1:50, ab983, Abcam, UK) or isotype control (1:50, ab91366, Abcam, UK) combined with Alexa Fluor 488-conjugated anti-mouse IgG (1:1000, #4408, Cell signaling, USA). However, the InsR protein expression in hepatocyte of C57 mice was evaluated by staining with Alexa Fluor 488-conjugated InsR monoclonal goat antibody (1:50, FAB1544G, R&D Systems, USA) or isotype control (1:50, IC108G, R&D Systems, USA). Then, these above protein expressions were analyzed by BD flow cytometer immediately. The excitation was at 494 nm for LDLR, 494 nm for InsR, 628 nm for p-AMPK, respectively. The emission was collected at 527 nm for LDLR and InsR, and 690 nm for p-AMPK, respectively.

**Immunofluorescence analysis.** Triple immunofluorescent staining of LDLR, InsR, and p-AMPK in hepG2 cells and liver tissue, was conducted by incubating simultaneously with a mouse InsR monoclonal antibody (1:50, sc-57342, Santa Cruz, USA), rabbit polyclonal antibody against p(Thr183/172)-AMPKα1/2 (1:100, YP0575, ImmunoWay, USA) and Alexa Fluor 488-conjugated rabbit LDLR antibody (1:100, ab196377, Abcam, UK). Then, samples were stained with CY3-conjugated goat anti-mouse secondary antibody (1:200, A10521, Thermofisher, USA) and Alexa Fluor 647-conjugated goat anti-rabbit secondary antibody (1:200, A32728, Thermofisher, USA). For double staining of IL-6 and TNF-α in liver and epididymal fat tissues, samples were stained with rabbit IL-6 (1:100, GB11117, Servicebio, China) and mouse TNF-α (1:100, GB11188, Servicebio, China) primary antibody and then incubated with FITC-conjugated goat anti-mouse secondary antibody (1:100, A16079, Thermo Fisher, USA) and CY3-conjugated goat anti-rabbit secondary antibody (1:100, A10520, Thermo Fisher, USA). Finally, all samples were mounted with DAPI. Fluorescence pictures were photographed on a C2t Nikon fluorescent microscope. The excitation at 377 nm was for nucleus, 494 nm for LDLR and TNF-α, 543 nm for InsR and IL-6, 628 nm for p-AMPK. The emission was collected at 447 nm for nucleus, 527 nm for LDLR and TNF-α, 586 nm for InsR and IL-6, 690 nm for p-AMPK.

**Data analysis and statistics**. Animals were excluded when an objective experimental failure was observed[55]. Values detected by the two folds of standard deviation were discarded[55]. The variance is similar between the groups that are being statistically compared. To avoid any systematic variation, Randomization was used in analysis. Studies were not blinded to investigators. The number of animals used in each study is listed in the figure legends. All repetitions of experiments were indicated in figure legends. The sample size in each experiment was chosen to ensure adequate power to detect a pre-specified effect.

Statistical analyses were performed in GraphPad Prism Software Version 5.0a (GraphPad, San Diego, CA) using two-sided Student's $t$-test. Normal distribution was tested using D'Agostino & Pearson's omnibus normality test. Data are presented as means ± standard error mean (SEM).

**Reporting summary**. Further information on research design is available in the Nature Research Reporting Summary linked to this article.

## Data availability

All data generated or analyzed during this study are included in this published article (and its supplementary information files). A reporting summary for this article is available as a Supplementary Information file. The data underlying Figs. 1c, 1d, 3b, 4b, 4c, 5b, 5c, 6a, 6b, 8a, 8d, 8e and Supplementary Figs. 2B, 4, 5 A, 5B, 6 C, 7A–D, 9, 12A–C, 13A–D, 14B, 15B–E, Supplementary Table 1 are provided as a Source Data file.

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

## Acknowledgements

This work was supported by the CAMS Innovation Fund for Medical Sciences (No. 2016-I2M-1–011, No. 2017-I2M-1–008, No. 2017-I2M-1–011, No. 2017-I2M-1–012, No. 2017-I2M-B&R-09); The Drug Innovation Major Project (No. 2018ZX09711001-003-002, 2018ZX09711001-002-005, 2018ZX09721003-009-007, 2018ZX09721003-009-003, 2018ZX09711001-011-003, 2017ZX09101003-003-002, 2016ZX09101017, 2015ZX09102-023-004); National Natural Science Foundation (No. 81621064); Science and Technology Program of Beijing (No. Z151100000115008). Beijing Key Laboratory (No. BZ0150, Z141102004414062).

## Author contributions

: H.H.G., C.L.F., W.X.Z., Z.G.L., H.J.Z., T.T.Z., C.M., Y.Z., R.L., S.W., Z.A., C.L., X.L.L., and X.L.M performed experiments and analytical methods. H.H.G., Z.G.L., W.X.Z., C.L. F., Y.X.H., W.S.Z., L.L.W., and J.D.J conceived and designed the experiments, interpreted and discussed the data, reviewed and edited the manuscript. L.L.W and J.D.J developed the hypothesis, coordinate the project and wrote the manuscript.
