## [Peer Review File · Nature Communications]

Reviewers' Comments:

Reviewer #1:

Remarks to the Author:

Comments to authors:

The manuscript by Guo and colleagues developed a smart micelle (BBR-CTA-Mic) consisting of a D- α -tocopheryl hydrophobic core and an on-site detachable cross-linked polyethylene glycol-thiol shell for effective liver deposition of BBR. They detected promoted expression of LDL-R, p-AMPK and Ins-R in hepG2 cells and in vivo. BBR-CTA-Mic oral gavage increased BBR liver accumulation and alleviated HFD induced hyperlipidemia, weight-gain and adiposity. This study provides promising method to increase the bioavailability and liver accumulation of BBR compared with previous approaches, showing potential interest. However, there are several concerns need to be addressed.

- 1) In Figure 1C-d, although BBR-CTA-Mic showed a robust BBR release in the exposure of 20% hepatic homogenate, however, compared with BBR-S or BBR-TPGS-Mic, BBR-CTA-Mic actually showed relatively low liver release of BBR. Please explain. It seems inconsistent to better liver uptake of BBR compared with BBR-S in Figure 2.
- 2) In Figure 1D, the author demonstrated that BBR-CTA-Mic showed better trans-epithelial drug transportation than other groups in vitro. What's the difference of the plasma amounts of BBR-CTA-Mic and BBR-TPGS-Mic after oral administration to demonstrate more BBR entering circulation through intestinal barrier?
- 3) In Figure 2, how about the comparison of intracellular BBR uptake between BBR-CTA-Mic and BBR-TPGS-Mic? This is important to show better effects of BBR-CTA-Mic in hepatic cells than BBR-TPGS-Mic. In addition, Figure 2A-d should be moved to after Figure 2B-d, corresponding to the same order appeared in the main text.
- 4) Figure 3, BBR-CTA-Mic treatment showed higher BBR intake in liver than other organs. Please discuss the possible mechanisms of liver target by BBR-CTA-Mic.
- 5) Figure 4 and Figure 5, the authors used InsR, LDL-R and p-AMPK as the readout of the effects of BBR-CTA-Mic by using confocal laser scanning microscopy, western, Flow cytometry and RT-PCR. RT-PCR is not appropriate to detect p-AMPK. Instead, the authors should detect total AMPK using western blotting. In addition to InsR expression, insulin signaling such as p-AKT should also be detected by western blotting. As different methods all demonstrate the expression of these proteins, I would suggest some data such as flow cytometry which is not commonly used for these protein detection, can be moved to supplement to condense the figures and improve readability.
- 6) Figure 5, HFD-fed mice were treated with BBR-S group (BS, 50mg kg⁻¹ of BBR), BBR-CTA-Mic group (BM, 50mg kg⁻¹ of BBR), and other indicated controls. Did the author compare BBR concentration in plasma, liver and adipose tissue among different groups in this HFD model? since they observe the effects of BBR in liver and adipose.
- 7) For Figure 5-Figure 8, the ideal control should be empty micelle group (EM) to exclude the nonspecific effects of micelle itself. Significance should be calculated between BBR-CTA-Mic and empty micelle group.
- 8) The phenotypes of BBR-CTA-Mic treatment in mouse models should be reorganized and condense the figures more logically. I would recommend put plasma parameters in Figure 5A, and body weight, tissue weight in Figure 6AB, and liver HE staining in Figure 7AB together as one figure. Molecular data such as Figure 5BCDE as a single figure. Put all the inflammation related data together as one Figure. Figure 8D can be moved right after Figure 8A to describe IF staining of TNF α and IL6 in liver and adipose tissue, same as the arrangement of western blotting and RT-PCR. Figure 6C could be moved to supplement. Again, Figures should appear in the same order to

the description in the main text.

9) Figure 7A-B, please also quantify liver TG content and red O oil staining.

10) One of the conclusion of the study is the potential effects of BBR-CTA-Mic on treating CMD. The authors compared oil-red staining images of aortic arches and diverging blood vessels in all groups in Figure S4. Please also the quantify this data to show the significance.

11) The writing of the manuscript should be improved to be more logic, clear and concise. As mentioned above, Figures should appear in the same order to the main text to make it easier to understood. For instance, according to the description in the results, Fig.S1BC should move after Fig. S1D. Also for Figure 2A-d, Figure 5, etc.

12) Line 339, Figure 3ABCD, each should be described clearly.

13) Line 405-428, Line 441-461, Line 500-510, Line 513-522 should move to discussion, to make the results section condensed and easier to read.

14) Figure labels using A-abcd are too complex. Some of the data can be condensed or moved to supplement.

Reviewer #2:

Remarks to the Author:

The MS by Guo et al on Liver-target Nanotechnology Facilitates Berberine to Ameliorate Cardio-metabolic Diseases is an extensive study on the positive effect of berberine micelles on various aspects of metabolic disease in the liver. My prime concern is the narrative of the study that the micelles passing intact through the intestine reach the plasma and ultimately release their cargo in the liver. Although tempting to speculate that this would be the case, it is notoriously difficult to stably encapsulate drugs within micelles in biological fluids and it is more likley that BBR separates form the micelles on its journey. The micelles might then still have a solubilizing/absorption enhancement effect which could still be valuable but the "smart liver delivery" as suggested by the authors may be less glamorous.

The studies performed do not confirm the narrative.

The release assays are performed in dialysis bags with 14 kD cutoff sizes, this effectively limits interaction with the majority of proteins, especially relevant in the serum (not plasma) incubations. As a result there is no acceptor for BBR inside the dialysis bag effectively limiting transfer. These studies, therefore only underline that the micelles are stable. Indeed when the dialysis-bag permeable GSH in liver homogenate is added and the micelles fall apart release is observed but this is not reflecting smart liver-specific behavior but rather the experimnetal set-up.

Similarly, the CaCo permeation of BBR which is in later experiments used as indication that the micelles pass intact over the monolayer, only measures BBR, so this will not tell in what form the BBR passed over the cell layer. Also the results of Rho123 permeation need to be interpreted with caution as interaction of Rho123 with lipidic molecules has been noted.

The subsequent step in the transport to the liver is difficult to imagine. Why would the micelle be able to avoid opsonization and interaction with any other cells type, but only chooses to engage with the hepatocyte? This is difficult to understand. These issues could be tackled with following the carrier as well as following the drug to see if they follow the same fate. Also competition experiments with other cells than the envisioned target cell could be informative to determine how specific the interaction is. Also a mechanism for the micelle behavior is needed. Which pathways (receptors and endocytic pathways) are responsible for micelle uptake by hepatocytes. Are MPS-

cells truly unable to internalize these constructs? In vivo, these studies should be complemented by a liver cell distribution profile to determine the specificity of the micelle for hepatocytes over Kupffer cells endothelial cells and stellate cells. Also, for example a study in apolipoprotein E knock-out mice could shed light on the distribution profile.

The biological effects are interesting and in line with successful delivery of large doses of BBR. Of course, to judge the benefit of micellar targeting it would be informative if a dose response (at least an increase of free BBR) could be added to see to what extent the micellar formulation truly changes the characteristics of the drug's performance beyond the 2.5-fold increase in liver uptake.

The language needs improvement as some synonyms are chosen that are not often used in the field which sometimes results in awkward sentences.

Reviewer #3:

Remarks to the Author:

The manuscript titled "Liver-target Nanotechnology Facilitates Berberine to Ameliorate Cardio-metabolic Diseases" by Guo et al. reports a method for the delivery of BBR to liver by wrapping BBR in BBR-loaded cross-link D- α -tocopheryl polyethylene glycol-thiol succinamide micelle. They claim this method effectively accumulates BBR in liver. By inducing the expression of LDL-R, p-AMPK and Ins-R, BBR-CTA-Mic administration ameliorates the metabolic disorders and attenuates aortic arch plaque formation.

The manuscript is clearly written and the results are well presented.

There are a few specific issues the authors should address by making modifications to the manuscript or by clarifying in their response, after which I would consider this work suitable for publication in Nature Communications.

Major comments:

Fig. 4C: a. It is shown that the InsR expression levels in the NC and BS groups are much lower than those in Fig. 5D. Please address why the InsR level is low here. b. Please test the level of phospho-InsR and evaluate the ratio of phospho-InsR to total InsR. c. Please test the expression level of AMPK and evaluate the ratio of phospho-AMPK to total AMPK.

Fig. 5D: a. Please test the level of phospho-InsR and evaluate the ratio of phospho-InsR to total InsR

b. Please test the expression of AMPK and evaluate the ratio of phospho-AMPK to total AMPK.

Fig. 6B: The adipocyte size in the BS group is larger than that in the NC and MC groups, shown in the panel on the left. However, as indicated in the BAR graph, the adipocyte size in the BS group is smaller than that in the MC group. Please address why there is the difference.

Minor comments:

Page 17, line 371: 'RT-PCR (Figure 4C) and western blot analysis (Figure 4D)' should be western blot analysis (Figure 4C) and RT-PCR (Figure 4D).

Fig. 5D: To be consistent with the labels in Fig 4C, "LDL-R and Ins-R" should be "LDLR and InsR".

Please improve the English by correcting general grammatical errors.

Reviewers' comments:

Reviewer #1 (Remarks to the Author):

Comments to authors:

The manuscript by Guo and colleagues developed a smart micelle (BBR-CTA-Mic) consisting of a D- α -tocopheryl hydrophobic core and an on-site detachable cross-linked polyethylene glycol-thiol shell for effective liver deposition of BBR. They detected promoted expression of LDL-R, p-AMPK and Ins-R in hepG2 cells and in vivo. BBR-CTA-Mic oral gavage increased BBR liver accumulation and alleviated HFD induced hyperlipidemia, weight-gain and adiposity. This study provides promising method to increase the bioavailability and liver accumulation of BBR compared with previous approaches, showing potential interest. However, there are several concerns need to be addressed.

Answer: Thank you very much for your hard work to review the article and valuable comments. In the past three months, we have done all the experiments you mentioned (see below), aiming for a good quality manuscript. To better address your concerns, we'd like to briefly introduce the design of the delivery system.

Berberine (BBR) is a lipid-lowering drug discovered by our group in 2004.¹ In the past decade, BBR has drawn increasing attention for its bioactivity of safely improving energy metabolism in patients with metabolic syndrome. For years, the challenging question came from BBR's unique pharmacokinetic characteristics, as the absolute bioavailability of BBR (oral) is very poor.^{2, 3} By principle, an active-site accumulation could be important for therapeutic reagents to execute their pharmacological effects, and liver is the main target of BBR.^{1, 4, 5} Therefore, we propose that a delivery system that can mediate a specific BBR liver-deposition might be a new strategy to enhance BBR's efficacy. Conventional micelles (such as TPGS-Mic) could increase the permeability and bioavailability of BBR.⁶ However; the use of TPGS-Mics is not successful, as they are unstable in the GI tract and circulation,

thus unable to take the drug for active-site accumulation.⁷ In this study, CTA-Mic was designed and developed to keep the advantages of the conventional TPGS micelles and increase the stability. As shown in the results, BBR-CTA-Mic could increase the accumulation of BBR in the liver, rather than in other organs (Fig. 3; the possible mechanisms are discussed in Q & A 4).

We consider the results of interesting for future formula optimization, aiming for a high efficient BBR treatment.

Q: 1) In Figure 1C-d, although BBR-CTA-Mic showed a robust BBR release in the exposure of 20% hepatic homogenate, however, compared with BBR-S or BBR-TPGS-Mic, BBR-CTA-Mic actually showed relatively low liver release of BBR. Please explain. It seems inconsistent to better liver uptake of BBR compared with BBR-S in Figure 2.

Answer: Yes, the Figure 1C contains confusion, and the description has been modified in the new version.

In fact, the Fig. 1C (a, -b, -c, -d) is made to show the release of BBR in the physiological environment after oral administration of BBR-CTA-Mic, which is in turn stomach, intestine, blood and liver. In the test, BBR-CTA-Mic showed a substantial release of BBR only in the liver homogenate tube. In this test, BBR-S (BBR solution) serves as the positive reference to show the maximal level of BBR in the test system and is higher than that of BBR-CTA-Mic; and the BBR-TPGS-Mic is used as a reference, showing less selectivity of BBR release in the abovementioned environment. In the revised version we have described this design in the Method section, to clarify the principle of the experiment. (Please see Line 621-630)

2) In Figure 1D, the author demonstrated that BBR-CTA-Mic showed better trans-epithelial drug transportation than other groups in vitro. What's the difference of the plasma amounts of BBR-CTA-Mic and BBR-TPGS-Mic after oral administration to demonstrate more BBR entering circulation through intestinal barrier?

Answer: As suggested, the plasma BBR level was compared between BBR-CTA-Mic (orally) and BBR-TPGS-Mic (orally). The result showed that the plasma amount of BBR is higher in the BBR-CTA-Mic treated mice than that in the BBR-TPGS-Mic treated ones. The new data has been included in the revised version (Please see page 332-335, Supplementary Fig. 8).

3) In Figure 2, how about the comparison of intracellular BBR uptake between BBR-CTA-Mic and BBR-TPGS-Mic? This is important to show better effects of BBR-CTA-Mic in hepatic cells than BBR-TPGS-Mic. In addition, Figure 2A-d should be moved to after Figure 2B-d, corresponding to the same order appeared in the main text.

Answer: We agree. In the revised version we have added the experiment results. BBR-CTA-Mic could increase the uptake of BBR, with an efficient higher than that of BBR-TPGS-Mic did. (Please see Line 274-279 and Supplement Fig. 4A, 4D, 4E)

Also, as suggested, in the new version, we rearranged the order of the figures to keep it consistent with order appearing in the manuscript. (Please see Fig. 2 and Supplement Fig. 4)

4) Figure3, BBR-CTA-Mic treatment showed higher BBR intake in liver than other organs. Please discuss the possible mechanisms of liver target by BBR-CTA-Mic.

Answer: After intestine absorption, liver is the first organ for BBR-CTA-Mic. The liver-accumulation property of BBR-CTA-Mic might be mediated through the following: 1) the new formula increased uptake of the entrapped BBR by hepatocyte; 2) the catabolized products of the vector in hepatocytes could inhibit the activity of CYP450s and P-gp efflux pump,⁸⁻¹⁰ thus slow-down the metabolism and efflux of BBR; and 3) the chemical and physical feature of the BBR-CTA-Mic (such as PEG chains on the surface of the carrier, as well as the diameter of the particles is within 20-100 nm arrange) could help it to evade the elimination by reticular-endothelial system in liver

(such as the Kupffer's cells).¹¹⁻¹³ In the revised version we have discussed the liver-accumulating mechanism of the BBR-CTA-Mic. (Please see Line 319-328)

5) Figure 4 and Figure 5, the authors used InsR, LDL-R and p-AMPK as the readout of the effects of BBR-CTA-Mic by using confocal laser scanning microscopy, western, Flow cytometry and RT-PCR. RT-PCR is not appropriate to detect p-AMPK. Instead, the authors should detect total AMPK using western blotting. In addition to InsR expression, insulin signaling such as p-AKT should also be detected by western blotting. As different methods all demonstrate the expression of these proteins, I would suggest some data such as flow cytometry which is not commonly used for these protein detection, can be moved to supplement to condense the figures and improve readability.

Answer: We agree. As suggested, we have added tests for the protein of InsR, *p*-InsR, LDLR, AMPK, *p*-AMPK, AKT and *p*-AMPK using Western blot, and evaluated the mRNA level of InsR, LDLR, AMPK and AKT using PCR, in HepG2 cells and in C57 mice. The results showed that the expression level of InsR, *p*-InsR, LDLR, AMPK and *p*-AMPK in HepG2 was increased after adding the BBR formulations; and the expression of InsR, *p*-InsR, LDLR, AMPK and *p*-AMPK, as well as AKT and p-AKT in mice treated with the BBR formulations was also elevated. We have added the results in revised manuscript. (Please see Fig. 4 and Fig. 5, Line342-355, 359-371)

As recommended, we have moved the flow cytometry data to the supplement information (Please see Supplement Fig. 9, 10)

Q: 6) Figure 5, HFD-fed mice were treated with BBR-S group (BS, 50mg kg⁻¹ of BBR), BBR-CTA-Mic group (BM, 50mg kg⁻¹ of BBR), and other indicated controls. Did the author compare BBR concentration in plasma, liver and adipose tissue among different groups in this HFD model? since they observe the effects of BBR in liver and adipose.

Answer: As suggested, we did the experiment. We have done the comparison of BBR concentration in the plasma, liver and adipose tissues among groups in the HFD feeding C57 mice model. The results showed that, BBR level in adipose of the BBR-CTA-Mic treated mice was higher than that of the BBR-S treated ones, but the increase of BBR in adipose by BBR-CTA-Mic is much less than that seen in liver tissues, suggesting its advantage of accumulating BBR in liver. The results are shown below. We did not add the figure in the revised version, because 1) the results were very similar to that done with normal mice (Fig. 3B), and 2) the manuscript already has 15 supplementary figures. However, the opinion of the reviewer will be fully respected.

Bio-distribution Evaluation HFD-fed C57BL/6J mice were administered with BBR-S or BBR-CTA-Mic (50 mg kg⁻¹ of BBR) *via* gavage injection. At each pre-determined time point, a group of five mice for each formulation were euthanized and blood (0.5ml) was obtained from posterior orbital venous plexus to a heparinized tube. The liver and fat tissues were harvested. The content of BBR in plasma, liver and adipose at different time points was achieved *via* LC/MS/MS. (n=5, mean ± SEM).

7) For Figure 5-Figure 8, the ideal control should be empty micelle group (EM) to exclude the nonspecific effects of micelle itself. Significance should be calculated between BBR-CTA-Mic and empty micelle group.

Answer: Agree with the suggestion, in the revised version, we have calculated the significance, comparing BBR-CTA-Mic with empty micelle group. (Please see Fig. 5, 6, 8)

8) The phenotypes of BBR-CTA-Mic treatment in mouse models should be reorganized and condense the figures more logically. I would recommend put plasma

parameters in Figure 5A, and body weight, tissue weight in Figure 6AB, and liver HE staining in Figure 7AB together as one figure. Molecular data such as Figure 5BCDE as a single figure. Put all the inflammation related data together as one Figure. Figure 8D can be moved right after Figure 8A to describe IF staining of TNFa and IL6 in liver and adipose tissue, same as the arrangement of western blotting and RT-PCR. Figure 6C could be moved to supplement. Again, Figures should appear in the same order to the description in the main text.

Answer: We agree. As suggested, we have reorganized and condensed the figures. In the revised version, we have put plasma parameters, body weight and tissue weight as one figure (Please see Fig. 6 in reversed version); we have put liver HE staining and liver Oil-red staining as one figure, (Please see Fig. 7 in reversed version); the molecular data have been put as a single figure (Please see Fig. 5 in reversed version); we have moved Fig. 8D right after Fig. 8A (Please see Fig. 8 in reversed version) and moved Fig. 6C to supplement. (Please see Supplement Fig. 12 in reversed version) We have double checked the Figures to make sure all the figures appear in the same order to the description in the main text. We believe, after the amendment suggested, the results in figures are presented more logically.

9) Figure 7A-B, please also quantify liver TG content and red O oil staining.

Answer: As instructed, we have quantified liver TG content and red O oil staining in the revised version, and the results have been added into the new version. (Please see Fig. 6A, Supplement Fig. 12B)

10) One of the conclusions of the study is the potential effects of BBR-CTA-Mic on treating CMD. The authors compared oil-red staining images of aortic arches and diverging blood vessels in all groups in Figure S4. Please also the quantity this data to show the significance.

Answer: Yes, we have quantified the oil-red staining using Image-Pro-Plus 6.0 software, in all groups. The results have been added in the revised version. (Please see Supplement Fig. 14)

11) The writing of the manuscript should be improved to be more logic, clear and concise. As mentioned above, Figures should appear in the same order to the main text to make it easier to understand. For instance, according to the description in the results, Fig.S1BC should move after Fig. S1D. Also for Figure 2A-d, Figure 5, etc.

Answer: In the revised manuscript, we have tried our best to improve the writing of the manuscript to make it more logic, clear and concise. We have double-checked the manuscript to make sure all the figures appear in the order identical to that in the text (Please see Fig. 2-8, Supplement Fig. 3-15). We have invited a native English speaker professional to proof-read the paper.

12) Line 339, Figure 3ABCD, each should be described clearly.

Answer: To clarify the Fig. 3, we have re-sentenced the description for Figure 3 and line 339 in the revised version. (Please see legend Fig. 3, Line305-313)

13) Line 405-428, Line 441-461, Line 500-510, Line 513-522 should move to discussion, to make the results section condensed and easier to read.

Answer: The manuscript is written in the way that the Results part is together with the Discussion, so that we could discuss issues right after the results presentation. However, if the reviewer considers that the Results part should be separated from the Discussion in this paper, we will do it.

14) Figure labels using A-abcd are too complex. Some of the data can be condensed or moved to supplement.

Answer: As suggested, in the revised version, we have moved some of the data to the supplement to condense the figure presentation. (Please see Fig. 2-8, Supplement Fig. 3-15)

Reference

- 1 Kong, W. *et al.* Berberine is a novel cholesterol-lowering drug working through a unique mechanism distinct from statins. *Nature medicine* **10**, 1344-1351, doi:10.1038/nm1135 (2004).
- 2 Gu S, *et al.* A metabolomic and pharmacokinetic study on the mechanism underlying the lipid-lowering effect of orally administered berberine. *Molecular bioSystems* **11**, 463-474 (2015).
3. Liu, C. S., Zheng, Y. R., Zhang, Y. F. & Long, X. Y. Research progress on berberine with a special focus on its oral bioavailability. *Fitoterapia* **109**, 274-282, doi:10.1016/j.fitote.2016.02.001 (2016).
- 4 Li, H. *et al.* Hepatocyte nuclear factor 1alpha plays a critical role in PCSK9 gene transcription and regulation by the natural hypocholesterolemic compound berberine. *The Journal of biological chemistry* **284**, 28885-28895, doi:10.1074/jbc.M109.052407 (2009).
- 5 Yuan, X. *et al.* Berberine ameliorates nonalcoholic fatty liver disease by a global modulation of hepatic mRNA and lncRNA expression profiles. *Journal of translational medicine* **13**, 24, doi:10.1186/s12967-015-0383-6 (2015).
6. Chen W, *et al.* Bioavailability study of berberine and the enhancing effects of TPGS on intestinal absorption in rats. *AAPS PharmSciTech* **12**, 705-711 (2011).
7. Sun T, Zhang YS, Pang B, Hyun DC, Yang M, Xia Y. Engineered nanoparticles for drug delivery in cancer therapy. *Angewandte Chemie (International ed in English)* **53**, 12320-12364 (2014).
8. Bogman K, Erne-Brand F, Alsenz J, Drewe J. The role of surfactants in the reversal of active transport mediated by multidrug resistance proteins. *Journal of pharmaceutical sciences* **92**, 1250-1261 (2003).
9. Rege BD, Kao JP, Polli JE. Effects of nonionic surfactants on membrane transporters in Caco-2 cell monolayers. *European journal of pharmaceutical sciences : official journal of the European Federation for Pharmaceutical Sciences* **16**, 237-246 (2002).
10. Wempe MF, *et al.* Inhibiting efflux with novel non-ionic surfactants: Rational design based on vitamin E TPGS. *International journal of pharmaceutics* **370**, 93-102 (2009).
11. Pasut G, *et al.* Polyethylene glycol (PEG)-dendron phospholipids as innovative constructs for the preparation of super stealth liposomes for anticancer therapy. *Journal of controlled release : official journal of the Controlled Release Society* **199**, 106-113 (2015).
12. Harris JM, Chess RB. Effect of pegylation on pharmaceuticals. *Nature reviews Drug discovery* **2**, 214-221 (2003).
13. Wang Y, *et al.* Novel galactosylated biodegradable nanoparticles for hepatocyte-delivery of oridonin. *International journal of pharmaceutics* **502**, 47-60 (2016).

Reviewer #2 (Remarks to the Author):

The MS by Guo et al on Liver-target Nanotechnology Facilitates Berberine to Ameliorate Cardio-metabolic Diseases is an extensive study on the positive effect of berberine micelles on various aspects of metabolic disease in the liver. My prime concern is the narrative of the study that the micelles passing intact through the intestine reach the plasma and ultimately release their cargo in the liver. Although tempting to speculate that this would be the case, it is notoriously difficult to stably encapsulate drugs within micelles in biological fluids and it is more likely that BBR separates from the micelles on its journey. The micelles might then still have a solubilizing/absorption enhancement effect which could still be valuable but the "smart liver delivery" as suggested by the authors may be less glamorous. The studies performed do not confirm the narrative.

Answer: Thank you very much for your hard work to review the manuscript and valuable suggestions, based on which, in the past 3 months, we have done most of the experiments you mentioned (see below). Furthermore, to better address your concerns, we'd like to briefly introduce the background of the study.

Berberine (BBR) is a lipid-lowering drug discovered by our group in 2004.¹ In the past decade, BBR has drawn increasing attention for its bioactivity of safely improving energy metabolism in patients with metabolic syndrome. For years, the challenging question came from BBR's unique pharmacokinetic characteristics, as the absolute bioavailability of BBR (oral) is very poor.^{2, 3} Whereas, an active-site accumulation could be crucial for therapeutic reagents to execute their pharmacological effects. Our data from *in vitro* experiments on hepatocyte^{1, 4-8}, and *in vivo* studies (including NOD/LtJ T1D and KK-Ay T2D mice,^{7, 9} hamsters,^{9, 10} as well as Sprague-Dawley (SD) rats^{1, 8, 11}) showed that is the main target of BBR.

Based on these, we propose that a delivery system that can mediate a selective BBR liver-deposition might be a new strategy to enhance BBR's efficacy. We agree with the point that conventional micelles (such as TPGS-Mic) could increase the

permeability and bioavailability of BBR; but the use of TPGS-Mics is not successful, as they are unstable and collapsed in the GI tract and circulation, thus unable to achieve their target organ in the intact form.^{12,13}

In this study, CTA-Mic was designed and developed to keep the advantages of the conventional TPGS micelles and increase the stability. In the novel system, ester bond in TPGS was replaced by a stronger amide bond to link D- α -Tocopherol succinate and PEG. Furthermore, through the forming of disulfide bond by the sulfhydryl at the end of PEG chain, a cross-linked outer shell was developed on the surface of the micelles. The outer shell, together with sturdy succinamide in the inner core, collaboratively contributes to the increased stability of BBR-CTA-Mics in physiological conditions. The particular intracellular environment of hepatocyte (such as high GSH and enzymes) could cause the collapse of the vector, leading to a burst release of loaded BBR. As shown in the results, BBR-CTA-Mic could 1) keep good stability when navigate in GI tract and circulation, and show a substantial release of BBR only in the hepatocyte environment (Fig. 1C, and Supplement Fig. 1); 2) increase penetration of BBR across intestinal epithelial wall with intact form (Fig. 1D, Supplement Fig. 3, please see answers to specific comments); and 3) accumulate BBR in the liver, rather than in other organs (Fig. 3). As liver is the first organ for BBR-CTA-Mic after intestine absorption, the fate of BBR-CTA-Mic in liver is critical for formula design. BBR-CTA-Mic showed more accumulation in the liver than that in other organs, probably because 1) the new formula increased the uptake of entrapped BBR by hepatocyte (see answers to specific comments); 2) the catabolized products of the vector in hepatocytes could inhibit the activity of CYP450s and P-gp efflux pump,¹⁴⁻¹⁶ thus slow-down the metabolism and efflux of BBR; and 3) the chemical and physical feature of the BBR-CTA-Mic (such as PEG chains on the surface of the carrier, as well as the diameter of the particles is within 20-100 nm arrange) could help it to evade the elimination by reticular-endothelial system in liver (such as the Kupffer's cells).¹⁷⁻¹⁹

We consider the results of interesting for future formula optimization, aiming for a high efficient BBR treatment.

1.

The release assays are performed in dialysis bags with 14 kD cutoff sizes, this effectively limits interaction with the majority of proteins, especially relevant in the serum (not plasma) incubations. As a result there is no acceptor for BBR inside the dialysis bag effectively limiting transfer. These studies, therefore only underline that the micelles are stable. Indeed when the dialysis-bag permeable GSH in liver homogenate is added and the micelles fall apart release is observed but this is not reflecting smart liver-specific behavior but rather the experimental set-up.

Answer:

We agree with the concern.

The goal of the assay was to test the organ-selective BBR release mediated by vectors. The four organ mimic environments tested in this project were stomach, intestine, blood and liver, representing the physiological steps after BBR-CTA-Mic oral administration.

The dialysis bag release test was designed to have BBR-CTA-Mic inside the bag and organ mimic environment outside the bag, in attempt to learn the release of BBR influenced by the organ environment. We thought that after exposure to the organ mimic environment (outside bag), BBR-CTA-Mic interacts with the organ components that have proper size to penetrate into the bag, and might cause structure change of the micelle, resulting in BBR release to the outside bag environment. We first started with 14KD bag, as it is often used in releasing test for formula investigation, and found that of the 4 organ mimic environments, liver environment is the one that caused a micelle structure change and the highest release of BBR (to outside bag). We did not go for large molecule cut-off bags, as the 14KD bag already showed selective release of BBR in liver. In the revised version we have described this design in the Method section, to clarify the principle of the experiment. (Please see Line 621-630) As the comment is so much valuable in the view of environment-micelle interaction (different molecule sizes) we will use dialysis bags with different cut-off in future investigation.

The liver-selective release of BBR-CTA-Mic is demonstrated by the *in vivo* organ distribution assay (Fig. 3), which showed that BBR-CTA-Mic increased BBR liver accumulation. The possible mechanisms are discussed in Q & A 3).

We have corrected “simulated plasma” to “simulated serum” in the revised version. (Please see Line 634 and Line 925)

2.

Similarly, the CaCo permeation of BBR which is in later experiments used as indication that the micelles pass intact over the monolayer, only measures BBR, so this will not tell in what form the BBR passed over the cell layer. Also the results of Rho123 permeation need to be interpreted with caution as interaction of Rho123 with lipidic molecules has been noted.

Answer:

As instructed, ultra-centrifugal filters (a new experiment) have been used to evaluate the integrity of micelles after the transportation across the monolayer, with a method described by Johnsen et al.²⁰ The result showed that, more than 50% of CTA-Mics keep intact across the monolayer, while almost all of the conventional TPGS-Mics collapsed during the transportation. (Please see Supplement Fig. 3, Line 232-236 and Line676-682)

We agree with the suggestion on the Rho123's results. Rho123 is a P-gp substrate and commonly used to investigate the activity of P-gp on drug transportation.²¹⁻²³ We have been very careful in the interpretation of the Rho123-related results.

3.

The subsequent step in the transport to the liver is difficult to imagine. Why would the micelle be able to avoid opsonization and interaction with any other cells type, but only chooses to engage with the hepatocyte? This is difficult to understand. These issues could be tackled with following the carrier as well as following the drug

to see if they follow the same fate. Also competition experiments with other cells than the envisioned target cell could be informative to determine how specific the interaction is. Also a mechanism for the micelle behavior is needed. Which pathways (receptors and endocytic pathways) are responsible for micelle uptake by hepatocytes. Are MPS-cells truly unable to internalize these constructs? In vivo, these studies should be complemented by a liver cell distribution profile to determine the specificity of the micelle for hepatocytes over Kupffer cells endothelial cells and stellate cells. Also, for example a study in apolipoprotein E knock-out mice could shed light on the distribution profile.

Answer:

We consider the question important to improve the quality of the manuscript, and have added several experiments to address the issues.

1) Liver- and cell-selectivity:

***As suggested**, we have added a new experiment to examine the mechanism of endocytosis of BBR-CTA-Mic in HepG2 cells. The result showed that clathrin-mediated endocytosis was the predominant one for BBR-CAT-Mic uptake, followed by caveolae-mediated endocytosis. (Please see Supplement Fig. 5, Line 279-282, 741-748)

***We also did new experiment** to evaluate the intracellular uptake of BBR in different cell lines, including HepG2 (liver), H446 (lung), 3T3 (adipose) and HT29 (colon). The results showed that, the BBR-CTA-Mics could improve BBR uptake in all of the cell lines, with no significant difference among cells.

Answer Fig. 1 BBR uptake in different cell lines. The cellular uptake of BBR-S and BBR-CTA-Mic by different cells was examined using qualitative confocal laser scanning microscopy (CLSM, up) and quantitative flow cytometry analysis. The cells were incubated with BBR-entrapped CTA-Mic or BBR-S for 4h.

***We have added new test** to study liver cell distribution profile of BBR-CTA-Mic *in vivo* by isolating hepatocytes, Kupffer cells, endothelial cells and stellate cells²⁴⁻²⁷, respectively, after oral administration of BBR-CTA-Mic. The result showed that, after reaching in liver tissues, the uptake of BBR-CTA-Mic by hepatocytes and Kupffer cells was significantly higher than that by endothelial cells and stellate cells. The uptake in hepatocytes appeared slightly high than that in Kupffer cells. (Peak shift) We have added these results in the revised manuscript. (Please see Line 328-332, Supplementary Fig. 7 and supplementary information)

***As suggested**, organ distribution of BBR formulations in ApoE knock-out mice has been done. Similar to the results in our study on C57 mice (Fig. 3 in manuscript), BBR-CTA-Mic increased the liver drug accumulation more than that in plasma and other organs. (Answer Fig. 2 showed below). For better clarity, we

did not show the results in the manuscript. However, if the reviewer considers the results essential, we will do so.

Answer Fig. 2 Bio-distribution assay. Apo E knockout mice were administered with BBR-S or BBR-CTA-Mic (50 mg kg^{-1} of BBR) *via* gavage injection. At each pre-determined time point, a group of five mice for each formulation were euthanized and blood (0.5ml) were obtained from posterior orbital venous plexus to a heparinized tube and major organs (heart, liver, spleen, lung and kidney) were harvested. The organ distribution of BBR was analyzed *via* LC/MS/MS.

2) Mechanism:

After intestine absorption, liver is the first organ for BBR-CTA-Mic after short travel from portal vein. The liver-accumulation of BBR-CTA-Mic could be explained by the following: 1) it could increase intestinal absorption and improve hepatocyte uptake of entrapped BBR; 2) degradation products of the vectors in liver cells could inhibit the activity of CYP450s and P-gp efflux, leading to a slow-down metabolism and secretion of BBR; 3) it could avoid the elimination by reticula-endothelial system in liver, including Kupffer's cells. This was verified by the *in vivo* distribution assay in our study. Our results showed that BBR-CTA-Mic accumulated in liver tissues, rather than plasma or other organs. We have added the mechanism explanation into the revised manuscript (Please see Line 319-328).

3) The fate of BBR and carrier *in vivo*.

We agree that chasing the carrier and the drug to see their fate *in vivo* is a very interesting idea. However, we realize that it is a big work, and therefore have had several meetings to discuss the possibility. The experiment might need to label raw materials with isotopes (or fluorescein) followed by chemical synthesis, thus other collaborator teams and administration permission (e.g. isotope) might be needed. Also, as the experiments might be quite labor- and time-consuming, we hope to have a chance to do it in the near future, if the reviewer agrees.

4.

The biological effects are interesting and in line with successful delivery of large doses of BBR. Of course, to judge the benefit of micellar targeting it would be informative if a dose response (at least an increase of free BBR) could be added to see to what extent the micellar formulation truly changes the characteristics of the drug's performance beyond the 2.5-fold increase in liver uptake.

Answer:

As suggested, BBR-CTA-Mic with gradient amount of BBR (low dose 25mg, middle dose 50mg and high dose 75mg kg⁻¹ day⁻¹ of BBR) were administered to C57BL/6J (6 weeks; 18-20 g) for two weeks by gavage. 4h after the last dose, the mice were anesthetized and the liver tissues were harvested. The gene expression of LDLR, InR and AMPK were tested using Western blot and RT-PCR. The BBR content in liver was also analyzed by LC-MS/MS. The result showed that, high dose BBR-CTA-Mic did show high BBR level in liver and was the most effective treatment for stimulating these genes in C57BL/6J mice. The gene inspiring effect was consistent with the BBR content in liver tissues. We have added these results in the revised manuscript. (Please see Supplement Fig. 11, Line 389-390, Supplement information section)

5.

The language needs improvement as some synonyms are chosen that are not often used in the field which sometimes results in awkward sentences.

Answer:

We agree. In the revised manuscript, we have tried our best to polish the English/grammar and then asked a native English speaker professional to proof read the paper. We hope that the revised manuscript meet the language criteria.

Reference

- 1 Kong, W. *et al.* Berberine is a novel cholesterol-lowering drug working through a unique mechanism distinct from statins. *Nature medicine* **10**, 1344-1351, doi:10.1038/nm1135 (2004).
- 2 Gu S, *et al.* A metabolomic and pharmacokinetic study on the mechanism underlying the lipid-lowering effect of orally administered berberine. *Molecular bioSystems* **11**, 463-474 (2015).
3. Liu, C. S., Zheng, Y. R., Zhang, Y. F. & Long, X. Y. Research progress on berberine with a special focus on its oral bioavailability. *Fitoterapia* **109**, 274-282, doi:10.1016/j.fitote.2016.02.001 (2016).
- 4 Kong WJ, *et al.* Combination of simvastatin with berberine improves the lipid-lowering efficacy. *Metabolism: clinical and experimental* **57**, 1029-1037 (2008).
- 5 Abidi P, Zhou Y, Jiang JD, Liu J. Extracellular signal-regulated kinase-dependent stabilization of hepatic low-density lipoprotein receptor mRNA by herbal medicine berberine. *Arteriosclerosis, thrombosis, and vascular biology* **25**, 2170-2176 (2005).
6. Li CH, Tang SC, Wong CH, Wang Y, Jiang JD, Chen Y. Berberine induces miR-373 expression in hepatocytes to inactivate hepatic steatosis associated AKT-S6 kinase pathway. *European journal of pharmacology* **825**, 107-118 (2018).
7. Kong WJ, *et al.* Berberine reduces insulin resistance through protein kinase C-dependent up-regulation of insulin receptor expression. *Metabolism: clinical and experimental* **58**, 109-119 (2009).
8. Li Z, Jiang JD, Kong WJ. Berberine up-regulates hepatic low-density lipoprotein receptor through Ras-independent but AMP-activated protein kinase-dependent Raf-1 activation. *Biological & pharmaceutical bulletin* **37**, 1766-1775 (2014).
9. Li XY, *et al.* Effect of Berberine on promoting the excretion of cholesterol in high-fat diet-induced hyperlipidemic hamsters. *Journal of translational medicine* **13**, 278 (2015).
10. Li Z, Geng YN, Jiang JD. Antioxidant and anti-inflammatory activities of berberine in the treatment of diabetes mellitus. **2014**, 289264 (2014).
11. Kong WJ, *et al.* Combination of simvastatin with berberine improves the lipid-lowering

- efficacy. *Metabolism: clinical and experimental* **57**, 1029-1037 (2008).
12. Chen W, *et al.* Bioavailability study of berberine and the enhancing effects of TPGS on intestinal absorption in rats. *AAPS PharmSciTech* **12**, 705-711 (2011).
 13. Sun T, Zhang YS, Pang B, Hyun DC, Yang M, Xia Y. Engineered nanoparticles for drug delivery in cancer therapy. *Angewandte Chemie (International ed in English)* **53**, 12320-12364 (2014).
 14. Bogman K, Erne-Brand F, Alsenz J, Drewe J. The role of surfactants in the reversal of active transport mediated by multidrug resistance proteins. *Journal of pharmaceutical sciences* **92**, 1250-1261 (2003).
 15. Rege BD, Kao JP, Polli JE. Effects of nonionic surfactants on membrane transporters in Caco-2 cell monolayers. *European journal of pharmaceutical sciences : official journal of the European Federation for Pharmaceutical Sciences* **16**, 237-246 (2002).
 16. Wempe MF, *et al.* Inhibiting efflux with novel non-ionic surfactants: Rational design based on vitamin E TPGS. *International journal of pharmaceutics* **370**, 93-102 (2009).
 17. Pasut G, *et al.* Polyethylene glycol (PEG)-dendron phospholipids as innovative constructs for the preparation of super stealth liposomes for anticancer therapy. *Journal of controlled release : official journal of the Controlled Release Society* **199**, 106-113 (2015).
 18. Harris JM, Chess RB. Effect of pegylation on pharmaceuticals. *Nature reviews Drug discovery* **2**, 214-221 (2003).
 19. Wang Y, *et al.* Novel galactosylated biodegradable nanoparticles for hepatocyte-delivery of oridonin. *International journal of pharmaceutics* **502**, 47-60 (2016).
 20. Johnsen E, *et al.* A critical evaluation of Amicon Ultra centrifugal filters for separating proteins, drugs and nanoparticles in biosamples. *J Pharmaceut Biomed* **120**, 106-111 (2016).
 21. Rege BD, Kao JP, Polli JE. Effects of nonionic surfactants on membrane transporters in Caco-2 cell monolayers. *European journal of pharmaceutical sciences : official journal of the European Federation for Pharmaceutical Sciences* **16**, 237-246 (2002).
 22. Sugihara N, Toyama K, Michihara A, Akasaki K, Tsuji H, Furuno K. Effect of benzo[a]pyrene on P-glycoprotein-mediated transport in Caco-2 cell monolayer. *Toxicology* **223**, 156-165 (2006).
 23. Xin HW, Tang X, Ouyang M, Zhong JX, Li WL. Effects of berberine on pharmacokinetics of midazolam and rhodamine 123 in rats in vivo. *SpringerPlus* **5**, 380 (2016).
 24. Bourgognon M, Klippstein R, Al-Jamal KT. Kupffer Cell Isolation for Nanoparticle Toxicity Testing. *Journal of Visualized Experiments*, (2015).
 25. Kegel V, Deharde D, Pfeiffer E, Zeilinger K, Seehofer D, Damm G. Protocol for Isolation of Primary Human Hepatocytes and Corresponding Major Populations of Non-parenchymal Liver Cells. *Journal of Visualized Experiments*, (2016).
 26. Cabral F, *et al.* Purification of Hepatocytes and Sinusoidal Endothelial Cells from Mouse Liver Perfusion. *Journal of Visualized Experiments*, (2018).
 27. Maschmeyer P, Flach M, Winau F. Seven steps to stellate cells. *Journal of visualized experiments : JoVE*, (2011).

Reviewer #3 (Remarks to the Author):

The manuscript titled “Liver-target Nanotechnology Facilitates Berberine to Ameliorate Cardio-metabolic Diseases” by Guo et al. reports a method for the delivery of BBR to liver by wrapping BBR in BBR-loaded cross-link D- α -tocopheryl polyethylene glycol-thiol succinamide micelle. They claim this method effectively accumulate BBR in liver. By inducing the expression of LDL-R, p-AMPK and Ins-R, BBR-CTA-Mic administration ameliorates the metabolic disorders and attenuates aortic arch plaque formation.

The manuscript is clearly written and the results are well presented.

There are a few specific issues the authors should address by making modifications to the manuscript or by clarifying in their response, after which I would consider this work suitable for publication in Nature Communications.

Answer: Thank you very much for your hard work to review the article and valuable comments.

Major comments:

Q4. Fig. 4C: a. It is shown that the InsR expression levels in the NC and BS groups are much lower than those in Fig. 5D . Please address why the InsR level is low here. b. Please test the level of phosphor-InsR and evaluate the ratio of phospho-InsR to total InsR. c. Please test the expression level of AMPK and evaluate the ratio of phospho-AMPK to total AMPK.

Answer:

1) In the Fig. 4, we tested gene expression in the human HepG2 liver cells, but in the Fig. 5, we tested the gene expression in liver tissues of C57 mice. The two different systems (cultured human cell vs mouse liver tissue) might be part of the reason; also, the reagents used in the test is different (e.g. antibody detection for human or mouse InsR).

2) We agree. As suggested, we have tested the level of phosphor-InsR and evaluated the ratio of phospho-InsR to total InsR, as well as tested the expression level of AMPK and evaluate the ratio of phospho-AMPK to total AMPK in the revised manuscript. (Please see Fig. 4 and Fig. 5, Line342-355, 359-371)

Q. Fig. 5D: a. Please test the level of phosphor-InsR and evaluate the ratio of phospho-InsR to total InsR

b. Please test the expression of AMPK and evaluate the ratio of phospho-AMPK to total AMPK.

Answer: Yes. As suggested, in the reversed version, we have tested the level of phosphor-InsR and evaluate the ratio of phospho-InsR to total InsR, as well as tested the level of phospho-AMPK and evaluate the ratio of phospho-AMPK to total AMPK *in vivo*. (Please see Fig. 4 and Fig. 5, Line342-355, 359-371)

Q: Fig. 6B: The adipocyte size in the BS group is larger than that in the NC and MC groups, shown in the panel on the left. However, as indicated in the BAR graph, the adipocyte size in the BS group is smaller than that in the MC group. Please address the why there is the difference.

Answer: We accept the criticism. In the original version, we misplaced the adipose pictures in Fig. 6. In the reversed version, we have corrected the mistake and double checked all the figures carefully. (Please see Fig. 6)

Minor comments:

Q: Page 17, line371: 'RT-PCR (Figure 4C) and western blot analysis (Figure 4D)' should be western blot analysis (Figure 4C) and RT-PCR (Figure 4D).

Answer: Thanks for the correction. We have corrected the sequence in describing the

methods, and checked it throughout the manuscript. (Please see Line 346, 348)

Fig. 5D: To be consistent with the labels in Fig 4C, “LDL-R and Ins-R” should be “LDLR and InsR”.

Answer: We have changed LDL-R and Ins-R in Fig. 4 to LDLR and InsR. We have double checked all the abbreviations in the manuscript to make sure the consistence.

Please improve the English by correcting general grammatical errors.

Answer: We have tried our best to polish the English/grammar and then asked a native English speaker professional to proof read the paper. We believe that the general grammatical error has been corrected and the writing has been improved following the reviewer’s instruction.

Reviewers' Comments:

Reviewer #1:

None

Reviewer #3:

Remarks to the Author:

I have looked at the response letter and the revised manuscript by Guo and colleagues, and found most of the points raised in the previous round of review have been satisfactorily addressed.

There is a minor issue the authors should address by clarifying in their response, after which I would consider this work suitable for publication in Nature Communications.

The minor issue is about Figure 4C. In the BBR-CTA-Mic treated group, we can see strong activation of InsR. However, the insulin signaling was not stimulated because phosphorylation of AKT was not induced.

Reviewer #4:

Remarks to the Author:

In the revised manuscript, the authors have addressed a number of concerns raised by Reviewer #2 during the initial review. The newly added experiments, such as using ultra-centrifugal filter to evaluate the integrity of the micelles after transportation across the Caco cell layer and dose-dependent effect of BBR-CTA-Mic on gene expression in the liver, are useful additions and address some of the technical concerns.

On the other hand, one major issue does remain. The authors claimed that BBR-CTA-Mic had preferential engagement in hepatocytes other than Kupffer cells, endothelial cells and stellate cells. This is in contrary to common observations. Liver sinusoidal endothelial cells (LSECs) and Kupffer cells are prominent cell types in liver for clearance of nanoparticles in blood circulation, and LSECs are the first cells encountered by nanoparticles when coming up from the portal vein (Ref: Park J-K, et al., Nanomedicine-NBM, 2016, 12, 1365). Then how does BBR-CTA-Mic escape from LSECs and Kupffer cells needs to be clarified. This is relevant to the scientific impact of the manuscript. The accuracy of the method used in Supplementary Fig. 7 is in doubt. A more reliable method is to stain the liver cell suspensions with cell markers for various cells and then calculate the double positive cell populations with BBR-CTA-Mic using flow cytometry.

Reviewers' comments:

Reviewer #3 (Remarks to the Author):

I have looked at the response letter and the revised manuscript by Guo and colleagues, and found most of the points raised in the previous round of review have been satisfactorily addressed.

There is a minor issue the authors should address by clarifying in their response, after which I would consider this work suitable for publication in Nature Communications.

The minor issue is about Figure 4C. In the BBR-CTA-Mic treated group, we can see strong activation of InsR. However, the insulin signaling was not stimulated because phosphorylation of AKT was not induced.

Answer: Thank you very much for your valuable comments.

We agree. Our result showed that BBR-CTA-Mic treatment activated InsR expression. However, the BBR formulations didn't increase the expression of AKT and p-AKT in the HepG2 cells cultured in insulin-free conventional medium. This phenomenon is consistent with previous findings that the AKT was activated by BBR only when insulin was present¹. We have added the discussion in the revised version (Please see Line 314-317).

Reference

1. Kong WJ, *et al.* Berberine reduces insulin resistance through protein kinase C-dependent up-regulation of insulin receptor expression. *Metabolism* **58**, 109-119 (2009).

Reviewer #4 (Remarks to the Author):

In the revised manuscript, the authors have addressed a number of concerns raised by Reviewer #2 during the initial review. The newly added experiments, such as using ultra-centrifugal filter to evaluate the integrity of the micelles after transportation across the Caco cell layer and dose-dependent effect of BBR-CTA-Mic on gene expression in the liver, are useful additions and address some of the technical concerns.

On the other hand, one major issue does remain. The authors claimed that BBR-CTA-Mic had preferential engagement in hepatocytes other than Kupffer cells, endothelial cells and stellate cells. This is in contrary to common observations. Liver sinusoidal endothelial cells (LSECs) and Kupffer cells are prominent cell types in liver for clearance of nanoparticles in blood circulation, and LSECs are the first cells encountered by nanoparticles when coming up from the portal vein (Ref: Park J-K, et al., Nanomedicine-NBM, 2016, 12, 1365). Then how does BBR-CTA-Mic escape from LSECs and Kupffer cells need to be clarified. This is relevant to the scientific impact of the manuscript. The accuracy of the method used in Supplementary Fig. 7 is in doubt. A more reliable method is to stain the liver cell suspensions with cell markers for various cells and then calculate the double positive cell populations with BBR-CTA-Mic using flow cytometry.

Answer: Thank you for your valuable suggestions. In the past three months, we have tried our best to detect BBR content in different cell types using the method you mentioned.

The result showed that the BBR content in the Kupffer cells of the mice treated with BBR-CTA-Mic was less than that from the mice treated with BBR solution (Supplementary Fig. 8A), indicating a reduced uptake / elimination of BBR by the kupffer cells in liver. Furthermore, LSECs showed almost no uptake of BBR, in both BBR solution and BBR-CTA-Mic-treated mice (Supplementary Fig. 8B). In the liver of

the BBR-CTA-Mic treated mice, the average BBR content in hepatocyte population was similar to that of Kupffer cells (Supplementary Fig. 8A-c). As the number of hepatocytes in liver is many times more than that of the other types of cells, the sum of BBR in total hepatocytes (mediated through the CTA-Mic entrapment) should be much more than that in other types of cells.

Regarding the possible mechanism, these results might be interpreted by: 1) The chemical and physical feature of the BBR-CTA-Mic (such as PEG chains on the surface of the carrier, as well as the diameter of the particles is within 20-100 nm arrange) could help it to evade the elimination by reticular-endothelial system in liver (such as the Kupffer's cells)¹⁻⁸; 2) The increased penetration and accumulation of BBR in hepatocytes was facilitated by CTA-Mics.

We did order NTCP antibody (PA5-80001, Thermofisher) for a better labeling of the hepatocytes in the liver cell suspension. However, we were informed recently that the antibody couldn't be available until March 2019. We consider that the new results in this revised version have addressed the concern. However, if the reviewer insists, we could conduct the experiment when we get the antibody.

We agree that, the expression of "BBR-CTA-Mic had preferential engagement in hepatocytes other than Kupffer cells, endothelial cells and stellate cells" might cause confusion, thus we have modified the expression in the new version. (Please see line 288-298 in revised text, Supplementary information and Supplementary Fig. 8).

Reference

1. Pasut G, *et al.* Polyethylene glycol (PEG)-dendron phospholipids as innovative constructs for the preparation of super stealth liposomes for anticancer therapy. *Journal of controlled release : official journal of the Controlled Release Society* **199**, 106-113 (2015).
2. Harris JM, Chess RB. Effect of pegylation on pharmaceuticals. *Nature reviews Drug discovery* **2**, 214-221 (2003).
3. Wang Y, *et al.* Novel galactosylated biodegradable nanoparticles for hepatocyte-delivery of oridonin. *International journal of pharmaceutics* **502**, 47-60 (2016).
4. Jin Xie, *et al.* Controlled PEGylation of Monodisperse Fe₃O₄ Nanoparticles for Reduced Non-Specific Uptake by Macrophage Cells. *Advanced Materials* 2007,19(20):3163 – 3166
5. Yang An Shu, *et al.* Serum Proteins Opsonization and Phagocytic Uptake of PEG-Modified PLGA Nanoparticles: Effect of Particle Size. 2012, *Advanced Materials Research* 1662-8985, Vols. 393-395, pp

6. Shann S Yu. Size- and charge-dependent non-specific uptake of PEGylated nanoparticles by macrophages. *Int J Nanomedicine*. 2012; 7: 799–813

7. Didier Bazile. Stealth Me. PEG-PLA Nanoparticles Avoid Uptake by the Mononuclear Phagocytes System. *Journal of Pharmaceutical Sciences*, Volume 84, Issue 4, April 1995, Pages 493-498

8. Younsoo Bae, Kazunori Kataoka. Intelligent polymeric micelles from functional poly(ethylene glycol)-poly(amino acid) block copolymers. *Advanced Drug Delivery Reviews*, Volume 61, Issue 10, 10 August 2009, Pages 768-784

Supplementary Fig. 8 Liver cell distribution after BBR-CTA-Mic or BBR-S treatment C57BL/6J mice were treated with BBR-CTA-Mic (50mg kg⁻¹ of BBR) or BBR-solution (50mg kg⁻¹ of BBR) by gavage, mice treated with PBS were used as control. Four hours after administration, the mice were anesthetized by an intraperitoneal injection of 30 mg kg⁻¹ pentobarbital. The liver tissues were harvest and liver cell suspension was obtained according to protocol. Then, the liver cell suspensions were stained with APC-conjugated rat F4/80 antibody (ab105155, Abcam, USA; or isotype control antibody 553988, BD, USA) for 30 min in order to evaluate BBR uptake in Kupffer cells an no-Kupffer cells (most of which are hepatocyte), or stained simultaneously with APC-conjugated rat F4/80 antibody and PE-conjugated rat CD14 (12-0141-82, ThermoFisher, USA) to analyze the uptake of BBR in endothelial cells. The BBR uptake in different cell lines was detected using flow cytometry. A. Drug uptake by kupffer cells and hepatocytes in liver tissue. A-a: Representative plots of Kupffer cells (F4/80⁺, up) and non-kupffer cells (most of which are hepatocytes, bottom), X-axis represents fluorecence of BBR. A-b: Histogram of BBR uptake by kupffer and hepatocytes; A-c: Mean fluorescent index ratio (MFI ratio vs control) of BBR in kupffer cells and hepatocyte. A. Drug uptake by endothelial cells and hepatocytes in liver tissue. B-a: Representative plots of endothelial cells (CD14⁺ F4/80⁻, bottom right) and hepatocyte (bottom left). B-b: Histogram of BBR uptake by endothelial cells and hepatocytes; B-c: Mean fluorescent index ratio (vs control) of BBR in kupffer cells and hepatocyte.

Reviewers' Comments:

Reviewer #3:

Remarks to the Author:

I feel that the point raised in the second round of review have been satisfactorily addressed, and suggest this work is suitable for publication in Nature Communications.

Reviewer #4:

Remarks to the Author:

In the revised manuscript, the authors did some work in analyzing the distribution of BBR and BBR-CTA-Mic in various cell types in the liver, including hepatocytes, Kupffer cells and liver sinusoidal endothelial cells (LSECs). However, CD14 is a marker for macrophages and many monocytes, but not for LSECs. Anti-CD146 should be used.

Reviewer #4 (Remarks to the Author):

In the revised manuscript, the authors did some work in analyzing the distribution of BBR and BBR-CTA-Mic in various cell types in the liver, including hepatocytes, Kupffer cells and liver sinusoidal endothelial cells (LSECs). However, CD14 is a marker for macrophages and many monocytes, but not for LSECs. Anti-CD146 should be used.

Answer: Thank you very much for your valuable suggestions. We think the comments are highly valuable for further understanding of the BBR-CTA-Mic mediated selective distribution of BBR described in the manuscript. As suggested, we have analyzed the distribution of BBR and BBR-CTA-Mic in various cell types in the liver of C57 mice, including hepatocytes, Kupffer cells and liver sinusoidal endothelial cells (LSECs). Our results showed that the content of BBR in hepatocytes was improved by the CTA-Mic system while the drug taken by liver nonparenchymal cells (such as kupffer cells, liver sinusoidal endothelial cells, monocytes, etc) was decreased or not significantly interfered by the delivery system. This effect benefits the therapeutic effect of active BBR. CD14 was used for LSECs staining in the experiments, as CD14 was reported to be a surface marker for the LSECs (ref 1-9). However, as kupffer cells are also positive for CD14, double-staining of the homogenized mice liver cell suspension (mainly containing hepatocytes, kupffer cells and LSECs) was done with both CD14 antibody as well as F4/80 antibody (for kupffer cell specific marker, identical to that used in the Supplementary Fig. 8A), in which the

CD14⁺F480⁻ cells could be considered as LSECs and the CD14⁺F4/80⁺ cells as kupffer cells. Thus, we could identify LSECs in the cell suspension (CD14⁺F480⁻ cells at the IV quadrant, the lower-right corner, see Supplementary Fig. 8Ba). Then, we analyzed the BBR level in the LSECs after gating the CD14⁺F480⁻ cells. As shown in the Supplementary Fig. 8Bb, the CD14⁺F4/80⁻ cells showed almost no uptake of BBR, in both BBR solution and BBR-CTA-Mic-treated mice.

We agree that CD146 might be a better marker for the LSECs, and have discussed that double staining analysis might also be needed as lymphocytes cells are positive for CD 146 (ref 10-12). In addition, we were told by the company that it will take some time to get the CD146 antibody into our lab after receiving our purchase order. Although we consider the presented result provided reasonable information to address the question, we are planning to do the suggested experiment in the near future, aiming for detailed cell category recognition for Mic-CTA-BBR uptake. We have added the information in the revised manuscript. (Please see Line 287-292 in main text and Line 92-97, 107-119 in supplementary information).

Reference:

1. Jersmann HP, et al. Synthesis and surface expression of CD14 by human endothelial cells. *Infection and immunity*, 2001.
2. Scoazec JY, et al. In situ immunophenotyping study of endothelial cells of the human hepatic sinusoid: results and functional implications. *Hepatology*, 1991.
3. Xu B, et al. Capillarization of hepatic sinusoid by liver endothelial cell-reactive autoantibodies in patients with cirrhosis and chronic hepatitis. *Am J Pathol*, 2003.

4. Lye W, et al. Isolation, culture and *in-vitro* growth pattern of hepatic sinusoidal endothelial cells. Chin J Biomed Eng, 2015.
5. Wang Z, et al. Isolation, Purification and Identification of Sinusoidal Endothelial Cells in Rat Liver. Henan Science, 2009.
6. Zhang D, et al. Research progress on relationship between hepatic sinusoidal endothelial cells and hepatocellular carcinoma. Journal of Lanzhou university (Medical Sciences) 2013.
7. Liu B, et al. An improved method for isolation, cultivation and identification of liver sinusoidal endothelial cells in mouse. Chinese journal of cell biology, 2009.
8. Huang Y, et al. Dynamic change of toll like receptor 4 and CD14 in acute hepatic failure by D-galactosamine. Chinese Journal of Microecology, 2007.
9. Zhu J, et al. Preliminary study of the isolation, cultivation and biological characteristics about liver sinusoidal endothelial cells in rats. Journal of digestive surgery, 2004.
10. Elshal MF, et al. CD146 (Mel-CAM), an adhesion marker of endothelial cells, is a novel marker of lymphocyte subset activation in normal peripheral blood. Blood, 2005.
11. Elshal MF, et al. A unique population of effector memory lymphocytes identified by CD146 having a distinct immunophenotypic and genomic profile". BMC Immunol, 2007.
12. Covas DT, et al. Multipotent mesenchymal stromal cells obtained from diverse human tissues share functional properties and gene-expression profile with CD146+ perivascular cells and fibroblasts". Exp. Hematol, 2007.

Reviewers' Comments:

Reviewer #4:

Remarks to the Author:

The authors have properly answered my concerns in the revised manuscript. I have no other questions for this work.